# Host-pathogen genetic interactions underlie tuberculosis susceptibility in genetically diverse mice

Clare M Smith[1,2]*, Richard E Baker[1], Megan K Proulx[1], Bibhuti B Mishra[1,3], Jarukit E Long[1], Sae Woong Park[4], Ha-Na Lee[4], Michael C Kiritsy[1], Michelle M Bellerose[1], Andrew J Olive[1], Kenan C Murphy[1], Kadamba Papavinasasundaram[1], Frederick J Boehm[1], Charlotte J Reames[1], Rachel K Meade[2,5], Brea K Hampton[6,7], Colton L Linnertz[6], Ginger D Shaw[6,8], Pablo Hock[6], Timothy A Bell[6], Sabine Ehrt[4], Dirk Schnappinger[4], Fernando Pardo-Manuel de Villena[6,8], Martin T Ferris[6], Thomas R Ioerger[9], Christopher M Sassetti[1]*

[1]Department of Microbiology and Physiological Systems, University of Massachusetts Medical School, Worcester, United States; [2]Department of Molecular Genetics and Microbiology, Duke University, Durham, United States; [3]Department of Immunology and Microbial Disease, Albany Medical College, Albany, United States; [4]Department of Microbiology and Immunology, Weill Cornell Medical College, New York, United States; [5]University Program in Genetics and Genomics, Duke University, Durham, United States; [6]Department of Genetics, University of North Carolina at Chapel Hill, Chapel Hill, United States; [7]Curriculum in Genetics and Molecular Biology, University of North Carolina at Chapel Hill, Chapel Hill, United States; [8]Lineberger Comprehensive Cancer Center, University of North Carolina at Chapel Hill, Chapel Hill, United States; [9]Department of Computer Science and Engineering, Texas A&M University, College Station, United States

*For correspondence:
clare.m.smith@duke.edu (CMS);
christopher.sassetti@umassmed.edu (CMS)

**Abstract:** The outcome of an encounter with *Mycobacterium tuberculosis* (*Mtb*) depends on the pathogen's ability to adapt to the variable immune pressures exerted by the host. Understanding this interplay has proven difficult, largely because experimentally tractable animal models do not recapitulate the heterogeneity of tuberculosis disease. We leveraged the genetically diverse Collaborative Cross (CC) mouse panel in conjunction with a library of *Mtb* mutants to create a resource for associating bacterial genetic requirements with host genetics and immunity. We report that CC strains vary dramatically in their susceptibility to infection and produce qualitatively distinct immune states. Global analysis of *Mtb* transposon mutant fitness (TnSeq) across the CC panel revealed that many virulence pathways are only required in specific host microenvironments, identifying a large fraction of the pathogen's genome that has been maintained to ensure fitness in a diverse population. Both immunological and bacterial traits can be associated with genetic variants distributed across the mouse genome, making the CC a unique population for identifying specific host-pathogen genetic interactions that influence pathogenesis.

## Editor's evaluation

This work takes advantage of the genetic diversity of a panel of mice, termed the collaborative cross, to identify those host factors that contribute to heterogeneous outcomes after tuberculosis infection. The authors infect this panel of mouse strains with pools of *Mycobacterium tuberculosis*

transposon mutants, allowing identification of specific host genotypes that confer fitness effects on certain bacterial mutants. The resulting analyses identify loci that affect quantitative immunological phenotypes or fitness of select bacterial mutants. The study is likely to be an important resource to microbiologists in general and those individuals studying the host immune response to tuberculosis infection

## Introduction

Infection with *Mycobacterium tuberculosis* (*Mtb*) produces heterogeneous outcomes that are influenced by genetic and phenotypic variation in both the host and the pathogen. Classic human genetic studies show that host variation influences immunity to tuberculosis (TB) (*Abel et al., 2018*; *Comstock, 1978*). Likewise, the co-evolution of *Mtb* with different populations across the globe has produced genetically distinct lineages that demonstrate variable virulence traits (*Gagneux et al., 2006*; *Hershberg et al., 2008*; *Wirth et al., 2008*). The role of genetic variation on each side of this interaction is established, yet the intimate evolutionary history of both genomes suggests that interactions between host and pathogen variants may represent an additional determinant of outcome (*McHenry et al., 2020*). Evidence for genetic interactions between host and pathogen genomes have been identified in several infections (*Ansari et al., 2017*; *Berthenet et al., 2018*), including TB (*Caws et al., 2008*; *Holt et al., 2018*; *Thuong et al., 2016*). However, the combinatorial complexity involved in identifying these relationships in natural populations have left the mechanisms largely unclear.

Mouse models have proven to be a powerful tool to understand mechanisms of susceptibility to TB. Host requirements for protective immunity were discovered by engineering mutations in the genome of standard laboratory strains of mice, such as C57BL/6 (B6), revealing a critical role of Th1 immunity. Mice lacking factors necessary for the production of Th1 cells or the protective cytokine interferon gamma (IFNγ) are profoundly susceptible to *Mtb* infection (*Caruso et al., 1999*; *Cooper et al., 1993*; *Cooper et al., 1997*; *Flynn et al., 1993*; *Saunders et al., 2002*). Defects in this same immune axis cause the human syndrome Mendelian Susceptibility to Mycobacterial Disease (MSMD) (*Altare et al., 1998*; *Bogunovic et al., 2012*; *Bustamante et al., 2014*; *Filipe-Santos et al., 2006*), demonstrating the value of knockout (KO) mice to characterize genetic variants of large effect. Similarly, the standard mouse model has been used to define *Mtb* genes that are specifically required for optimal bacterial fitness during infection (*Bellerose et al., 2020*; *Sassetti and Rubin, 2003*; *Zhang et al., 2013*).

Despite the utility of standard mouse models, it has become increasingly clear that the immune response to *Mtb* in genetically diverse populations is more heterogeneous than any single small animal model (*Smith and Sassetti, 2018*). For example, while IFNγ-producing T cells are critical for protective immunity in standard inbred lines of mice, a significant fraction of humans exposed to *Mtb* control the infection without producing a durable IFNγ response (*Lu et al., 2019*). Similarly, IL-17 producing T cells have been implicated in both protective responses and inflammatory tissue damage in TB, but IL-17 has little effect on disease progression in B6 mice, except in the context of vaccination or infection with particularly virulent *Mtb* (*Gopal et al., 2012*; *Khader et al., 2007*). The immunological homogeneity of standard mouse models may also explain why only a small minority of the >4000 genes that have been retained in the genome of *Mtb* during its natural history promote fitness in the mouse (*Bellerose et al., 2020*). Thus, homogenous mouse models of TB fail to capture the distinct disease states, mechanisms of protective immunity, and selective pressures on the bacterium that are observed in natural populations.

The Collaborative Cross (CC) and Diversity Outbred (DO) mouse populations are new mammalian resources that more accurately represent the genetic and phenotypic heterogeneity observed in outbred populations (*Churchill et al., 2004*; *Churchill et al., 2012*). These mouse panels are both derived from the same eight diverse founder strains but have distinct population structures (*Saul et al., 2019*). DO mice are maintained as an outbred population and each animal represents a unique and largely heterozygous genome (*Keller et al., 2018*; *Svenson et al., 2012*). In contrast, each inbred CC strain's genome is almost entirely homozygous, producing a genetically stable and reproducible population in which the phenotypic effect of recessive mutations is maximized (*Shorter et al., 2019*; *Srivastava et al., 2017*). Together, these resources have been leveraged to identify host loci underlying the immune response to infectious diseases (*Noll et al., 2019*). In the context of TB, DO mice have been used as individual, unique hosts to identify correlates of disease, which resemble those

observed in non-human primates and humans (*Ahmed et al., 2020*; *Gopal et al., 2013*; *Koyuncu et al., 2021*; *Niazi et al., 2015*). Small panels of the reproducible CC strains have been leveraged to identify host background as a determinant of the protective efficacy of BCG vaccination (*Smith et al., 2016*) and a specific variant underlying protective immunity to tuberculosis (*Smith et al., 2019*). While these studies demonstrate the tractability of the DO and CC populations to model the influence of host diversity on infection, dissecting host-pathogen interactions requires the integration of pathogen genetic diversity.

We combined the natural but reproducible host variation of the CC panel with a comprehensive library of *Mtb* transposon mutants to determine whether the CC population could be used to characterize the interactions between host and pathogen. Using over 60 diverse mouse strains, we report that the CC panel encompasses a broad spectrum of TB susceptibility and immune phenotypes. By leveraging high-resolution bacterial phenotyping known as 'Transposon Sequencing' (TnSeq), we quantified the relative fitness of a saturated library of *Mtb* mutants across the CC panel and specific immunological mouse knockout strains. We report that approximately three times more bacterial genes contribute to fitness in the diverse panel than in any single mouse strain, defining a large fraction of the bacterial genome that is dedicated to adapting to distinct immune states. Association of both host immunological phenotypes and bacterial fitness traits with Quantitative Trait Loci (QTL) demonstrated the presence of discrete Host-Interacting-with Pathogen QTL (*Hip*QTL) that represent inter-species genetic interactions that influence the pathogenesis of this infection. Together, these observations support the CC population as a tractable model of host diversity that greatly expands the spectrum of immunological and pathological states that can be modeled in the mouse.

## Results

### The spectrum of TB disease traits in the CC exceeds that observed in standard inbred mice

To characterize the diversity of disease states that are possible in a genetically diverse mouse population, we infected a panel of 52 CC lines and the eight founder strains with *Mtb*. To enable bacterial transposon sequencing (TnSeq) studies downstream, the animals were infected via the intravenous (IV) route with a saturated library of *Mtb* transposon mutants (infectious dose of $10^5$ CFU), which in sum produce an infection that is similar to the wild-type parental strain (*Bellerose et al., 2020*; *Sassetti and Rubin, 2003*). Groups of three to six male mice per genotype were infected and TB disease-related traits were quantified at one-month post-infection. Data from all surviving animals that were phenotyped are provided in *Figure 1—source data 1*. The bacterial burden after 4 weeks of infection was assessed by plating (colony-forming units, CFU) and quantifying the number of bacterial chromosomes in the tissue (chromosome equivalents, CEQ). These two metrics were highly correlated ($r = 0.88$) and revealed a wide variation in bacterial burden across the panel (*Figure 1A* and *Figure 1—figure supplement 1*). The phenotypes of the inbred founder strains were largely consistent with previous studies employing an aerosol infection (*Smith et al., 2016*), where the WSB strain was more susceptible than the more standard B6, 129S1/SvlmJ (129), and NOD/ShiLtJ (NOD) strains. Across the entire CC panel, lung bacterial burden varied by more than 1000-fold, ranging from animals that are significantly more resistant than B6, to mice that harbored more than $10^9$ bacteria in their lungs (*Figure 1A*). Bacterial burden in the spleen also varied several thousand-fold across the panel and was moderately correlated with lung burden ($r = 0.43$) (*Figure 1—figure supplement 1*). Thus, the CC panel encompasses a large quantitative range of susceptibility.

Comparing various measures of infection progression showed many expected correlations but also an unexpected decoupling of some phenotypes. As an initial assessment of the disease processes in these animals, we correlated bacterial burden and lung cytokine abundance with measures of systemic disease such as weight loss and sufficient morbidity to require euthanasia ('earliness of death'). In general, correlations between these metrics indicated that systemic disease was associated with bacterial replication and inflammation (*Figure 1B* and *Figure 1—figure supplement 1*). Lung CFU was strongly correlated with weight loss, mediators that enhance neutrophil differentiation or migration (CXCL2 (MIP-2; $r = 0.79$), CCL3 (MIP-1a; $r = 0.77$), G-CSF ($r = 0.78$), and CXCL1 (KC; $r = 0.76$)), and more general proinflammatory cytokines (IL-6 ($r = 0.80$) and IL-1α ($r = 0.76$)) (*Figure 1—figure supplement 1*). These findings are consistent with previous work in the DO panel, that found both

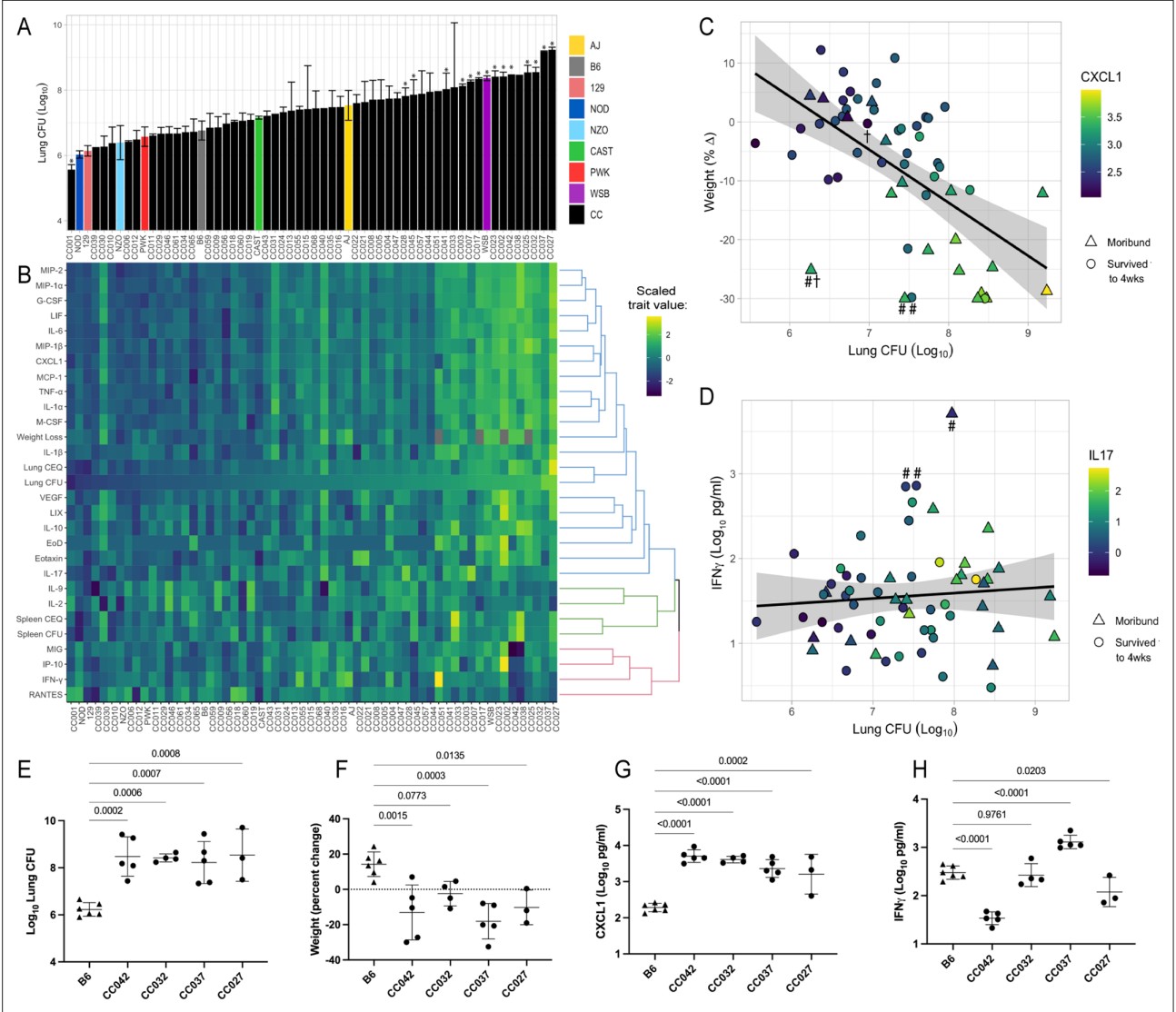

**Figure 1.** he spectrum of *M.tuberculosis* disease-related traits across the collaborative cross. (**A**) Average lung CFU (log$_{10}$) across the CC panel at 4 weeks post-infection. Bars show mean ± SD for CFU per CC or parental strain; groups of three to six mice per genotype were infected via IV route (infectious dose of $10^4$ in the lungs and $10^5$ in the spleen as quantified by plating CFU 24 hr post-infection). To compare the field standard B6 mouse strain with the diverse CC mouse strains, bars noted with * indicate strains that were statistically different from B6 (p < 0.05; 1-factor ANOVA with Dunnett's post-test). (**B**) Heatmap of the 32 disease-related traits (log$_{10}$ transformed) measured including: lung and spleen colony forming units (CFU); lung and spleen chromosomal equivalents (CEQ); weight loss (% change); cytokines from lung; 'earliness of death' (EoD), reflecting the number of days prior to the end of experiment that moribund strains were euthanized. Mouse genotypes are ordered by lung CFU. Scaled trait values were clustered (*hclust* in R package *heatmaply*) and dendrogram nodes colored by 3 k-means. Blue node reflects correlation coefficient R > 0.7; green R = 0.3–0.6 and red R < 0.2. Source files of all measured phenotypes are available in *Figure 1—source data 1*. (**C**) Correlation of lung CFU and weight (% change) shaded by CXCL1 levels. Genotypes identified as statistical outliers for weight are noted by #; CXCL1 by † (CC030 is triangle with #†;CC040 is triangle with #; AJ is circle with #; CC056 is circle with †). (**D**) Correlation of lung CFU and IFNγ levels shaded by IL-17. Strains identified as outliers for IFNγ noted by # (CC055 is left circle with #, AJ is right circle with #, CC051 is triangle with #). Each point in (**C**) and (**D**) is the average value per genotype. Outlier genotypes were identified after linear regression using studentized residuals. (**E–H**) Disease traits measured in a validation cohort (B6 vs CC042, CC032, CC037, and CC027) at 4 weeks after post low-dose aerosol infection (**E**) lung CFU (log$_{10}$); (**F**) Weight (percent change relative to uninfected); (**G**) CXCL1 abundance in lung (log$_{10}$ pg/mL homogenate); (**H**) IFNγ (log$_{10}$ pg/mL homogenate). Bar plots show the mean ± SD. p-Values indicate strains that were statistically different from B6 (1-factor ANOVA with Dunnett's post-test). Source files of all measured phenotypes in the aerosol validation cohort are available in *Figure 1—source data 2*. Groups consist of three to six mice per genotype. All mice in the initial CC screen and validation cohort were male.

The online version of this article includes the following source data and figure supplement(s) for figure 1:

**Source data 1.** CC TB disease phenotypes.

*Figure 1 continued on next page*

*Figure 1 continued*

**Source data 2.** Aerosol validation phenotypes.

**Figure supplement 1.** Phenotypic relationships between TB disease-related traits in the CC IV screen.

**Figure supplement 2.** Phenotypic relationships between TB disease-related traits in the aerosol validation cohort.

proinflammatory chemokines and neutrophil accumulation to be predictors of disease (*Ahmed et al., 2020*; *Gopal et al., 2013*; *Koyuncu et al., 2021*; *Niazi et al., 2015*).

The reproducibility of CC genotypes allowed us to quantitatively assess the heritability ($h^2$) of these immunological and disease traits. The percent of the variation attributed to genotype ranged from 56%–87% (mean = 73.4%; (*Appendix 1—table 1*)). The dominant role of genetic background in determining the observed phenotypic range allowed a more rigorous assessment of strains possessing outlier phenotypes than is possible in the DO population, based on linear regression using studentized residuals that accounts for the intragenotype variation. For example, despite the correlation between lung CFU and weight loss ($r = 0.57$), several strains failed to conform to this relationship (*Figure 1C*). In particular, CC030/GeniUnc ($p = 0.003$), CC040/TauUnc ($p = 0.027$) and A/J ($p = 0.03$) lost more weight than their bacterial burdens would predict (*Figure 1C*; noted by #). Similarly, CXCL1 abundance was higher in CC030/GeniUnc ($p = 0.001$) and lower in CC056/GeniUnc ($p = 0.040$), than the level predicted by their respective bacterial burden (*Figure 1C*; outlier genotypes noted by †). Thus, these related disease traits can be dissociated based on the genetic composition of the host.

The cluster of cytokines that was most notably unrelated to bacterial burden included IFNγ and the interferon-inducible chemokines CXCL10 (IP10), CXCL9 (MIG), and CCL5 (RANTES) (Red cluster in *Figure 1B*; *Figure 1—figure supplement 1*) ($R < 0.3$). Despite the clear protective role for IFNγ (*Cooper et al., 1993*; *Flynn et al., 1993*), high levels have been observed in susceptible mice, likely as a result of high antigen load (*Barber et al., 2011*; *Lazar-Molnar et al., 2010*). While high IFNγ levels in susceptible animals was therefore expected, it was more surprising to find a number of genotypes that were able to control bacterial replication yet had very low levels of this critically important cytokine (*Figure 1D*). This observation is likely due the inclusion of two founder lines, CAST/EiJ (CAST) and PWK/PhJ (PWK) that we previously found to display this unusual phenotype (*Smith et al., 2016*).

To assess the reproducibility of these findings in an aerosol infection model, we tested four CC genotypes that were susceptible by IV infection, including CC027, CC032, CC037, and CC042. We infected groups of 4–6 mice per genotype with H37Rv strain via low-dose aerosol infection (~100 CFU), including B6 mice as resistant controls. At 4-weeks post infection, we quantified lung CFU, lung cytokine abundance and weight loss as measurements of TB disease. Compared to the resistant B6 mice, the selected CC strains demonstrated higher bacterial burden in the lung (*Figure 1E*) and

**Table 1.** Disease-related Tuberculosis ImmunoPhenotype QTL (*Tip*QTL).
Multiple QTL within the same interval and clear allele effects are designated with the same *Tip*QTL number. p-Values are determined by Churchill-Doerge permutations (*Churchill and Doerge, 1994*). Column headings: QTL, quantitative trait loci; Chr, chromosome; LOD, logarithm of the odds; CEQ, chromosomal equivalents.

| QTL | Trait | Chr | LOD | p value | Interval start (Mb) | Peak (Mb) | Interval end (Mb) |
|------|-----------|-----|------|----------|--------------------|-----------|-------------------|
| *Tip5* | Spleen CEQ | 2 | 9.14 | 2.38E-02 | 174.29 | 178.25 | 178.25 |
| *Tip5* | Spleen CFU | 2 | 7.04 | 2.19E-01 | 73.98 | 174.29 | 180.10 |
| *Tip6* | IL-9 | 2 | 8.61 | 4.52E-02 | 33.43 | 41.4 | 41.48 |
| *Tip6* | IL-9 | 2 | 7.85 | 1.26E-01 | 22.77 | 24.62 | 25.65 |
| *Tip7* | IL-17 | 15 | 7.84 | 5.27E-02 | 67.98 | 74.14 | 82.11 |
| *Tip8* | CXCL1 | 7 | 7.57 | 1.06E-01 | 30.43 | 45.22 | 46.72 |
| *Tip8* | Lung CFU | 7 | 7.47 | 1.17E-01 | 31.06 | 37.78 | 45.22 |
| *Tip9* | IL-10 | 17 | 7.16 | 1.85E-01 | 80.98 | 82.47 | 83.55 |
| *Tip10* | Lung CFU | 15 | 7.13 | 1.86E-01 | 77.00 | 78.16 | 78.70 |

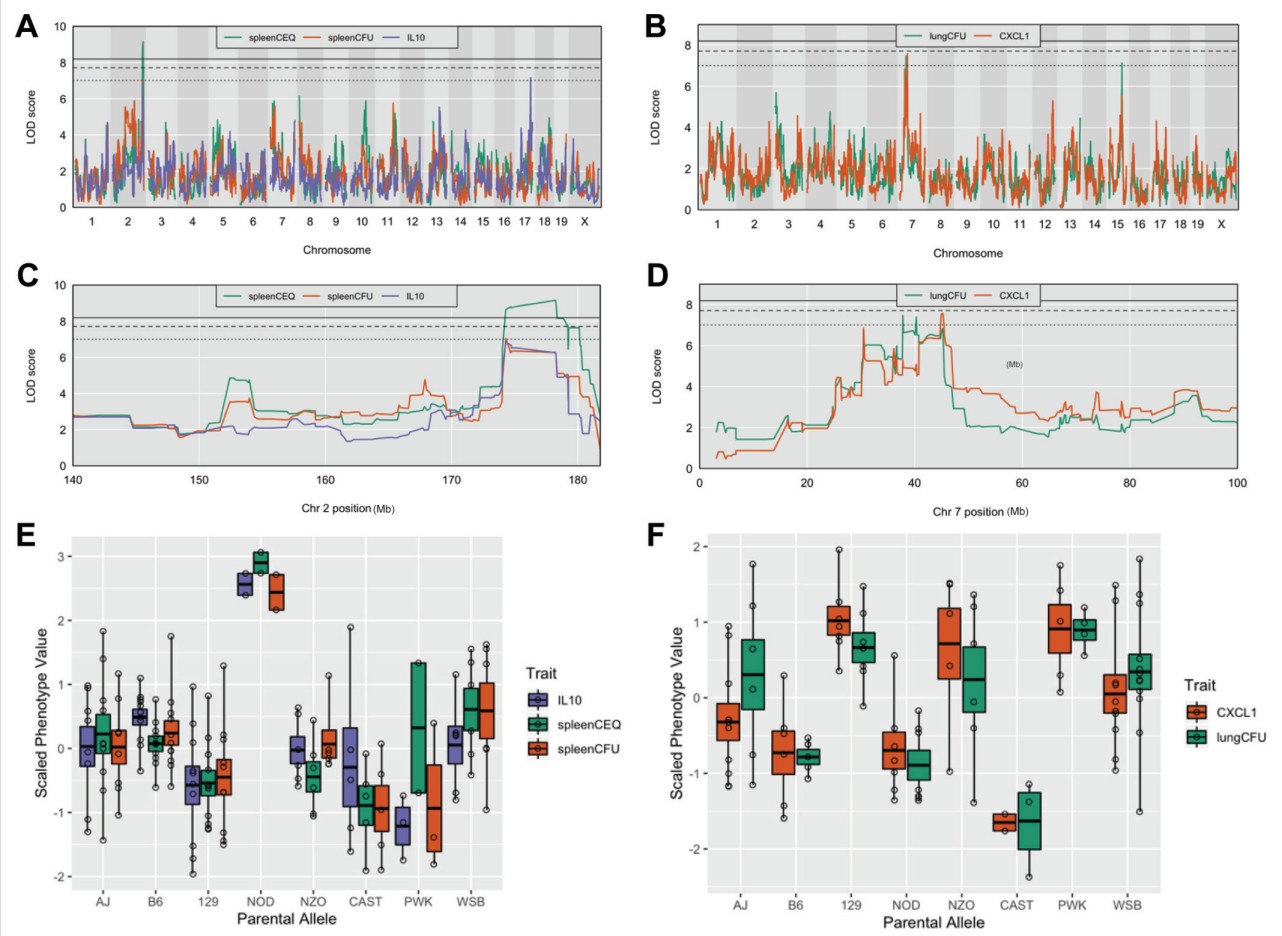

**Figure 2.** Host loci underlying TB disease-related traits. (**A–B**) Whole genome QTL scans of (**A**) spleen CEQ, spleen CFU and IL-10 (**B**) lung CFU and CXCL1. (**C**) Zoom of chromosome two loci. (**D**) Zoom of chromosome seven loci. Thresholds were determined by permutation analysis; solid line, middle dashed line, and lowest dotted lines represent p = 0.05, p = 0.1, and p = 0.2. (**E–F**) Scaled phenotype value per haplotype at the QTL peak marker. Each dot represents the mean value for a genotype.

significant weight loss (*Figure 1F*), thus validating disease traits as consistent across both route and dose. Likewise, cytokines that were highly correlated with lung burden in the CC screen (*Figure 1B*, *Figure 1—figure supplement 1*) were consistent in the aerosol validation study (*Figure 1—figure supplement 2*). Notably, CXCL1 was consistently high in the susceptible genotypes, as compared to B6 (*Figure 1G*), and was highly correlated with lung burden by both IV (*R* = 0.76) and aerosol (*R* = 0.92) routes. IFNγ levels were variable across the strains (*Figure 1H*) and did not correlate with lung CFU (*R* = −0.22), concordant with findings from the CC screen (*R* = −0.21). Altogether, this survey of TB-related traits demonstrated a broad range of susceptibility and the presence of qualitatively distinct and genetically determined disease states.

### *Tip*QTL define genetic variants that control TB immunophenotypes

Tuberculosis ImmunoPhenotype Quantitative Trait Loci (*Tip*QTL), which were associated with TB disease or cytokine traits, were identified and numbered in accordance with previously reported *Tip*QTL (*Smith et al., 2019*). Of the 32 TB-disease traits, we identified nine individual metrics that were associated with a chromosomal locus. Of these, three were associated with high confidence (p ≤ 0.053), and six other QTL met a suggestive threshold as determined by permutation analysis (p < 0.2; *Table 1*). Several individual trait QTL occupied the same chromosomal locations. For example, spleen CFU and spleen CEQ, which are both measures of bacterial burden and highly correlated, were associated with the same interval on distal chromosome 2 (*Table 1*, *Tip5*; *Figure 2A and C*). IL-10 abundance was associated with two distinct QTL (*Table 1*). While IL-10 was only moderately

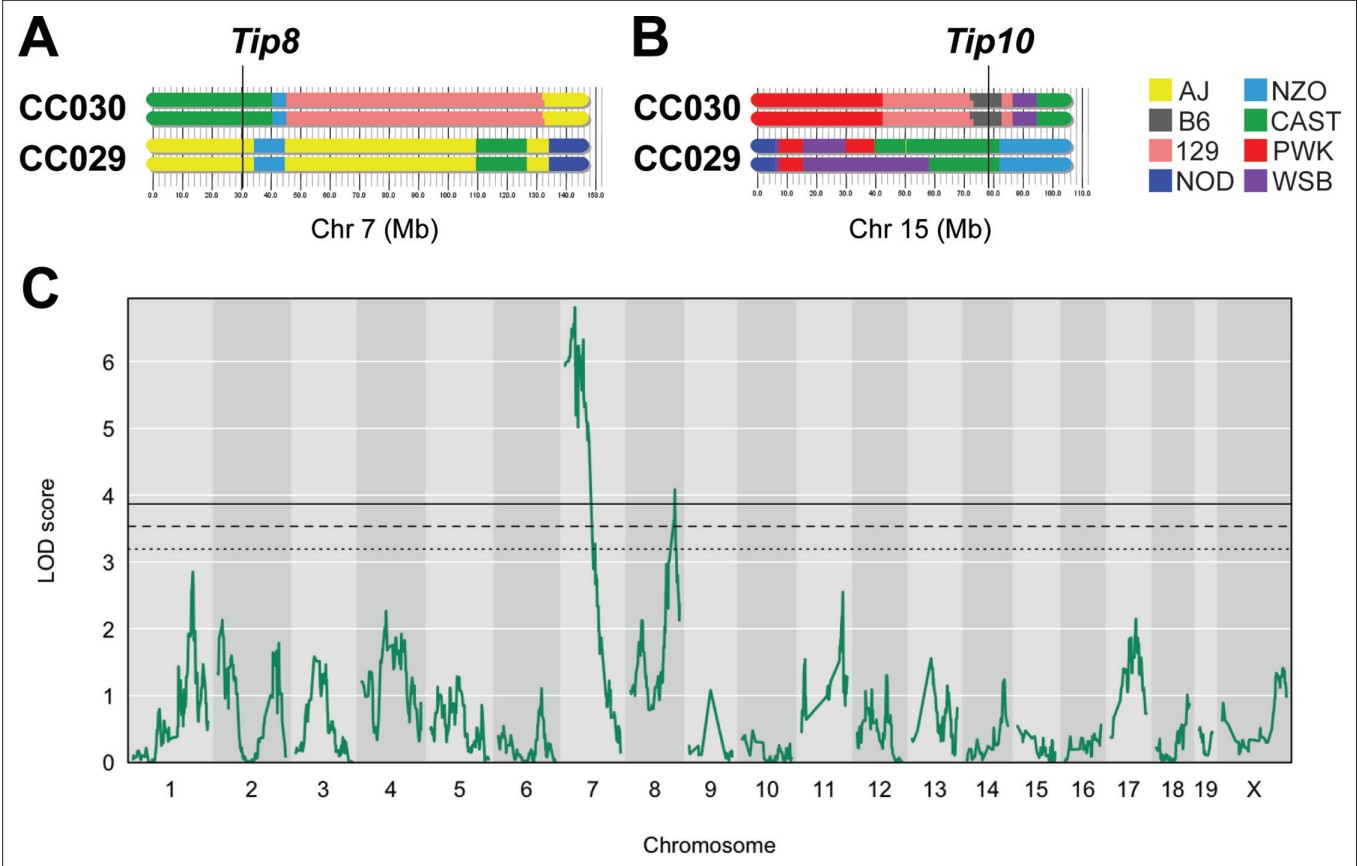

**Figure 3.** An F$_2$ intercross approach to validate QTL underlying lung CFU. (**A**) Haplotypes of CC030 and CC029 CC strains at Chr7 (*Tip 8*) and (**B**) at Chr15 (*Tip10*). The F$_2$ population (n = 251) based on these founders were genotyped, infected with *Mtb* (10$^5$ infectious dose by IV route, as per the original CC screen), and lung CFU was quantified at 1 month post-infection. (**C**) QTL mapping identified genome-wide significant (p < 0.05) loci on Chr7 (LOD = 6.81; peak position on Chr7 at 28.6 Mb) overlapping with *Tip8* and a new locus on Chr8 (LOD = 4.08; peak position Ch8:116.1 Mb). Thresholds were determined by permutation analysis; solid line, middle dashed line, and lowest dotted lines represent p = 0.05, p = 0.1, and p = 0.2. Source files of F$_2$ genotypes are available in *Figure 3—source data 1*; phenotypes are available in *Figure 3—source data 1*.

The online version of this article includes the following source data for figure 3:

**Source data 1.** F$_2$ Intercross genotype data.

**Source data 2.** F$_2$ Intercross phenotype data.

correlated with spleen CFU (*R* = 0.48), one of its QTL fell within the *Tip5* bacterial burden interval on chromosome 2 (*Figure 2A and C*). At this QTL, the NOD haplotype was associated with high values for all three traits (*Figure 2E*). Similarly, the strongly correlated traits, CXCL1 abundance and lung CFU, were individually associated to the same region on chromosome 7 (*Table 1, Tip8*; *Figure 2B and D*). In this interval, the CAST haplotype was associated with both low bacterial burden and CXCL1 (*Figure 2F*). At both *Tip5* and *Tip8*, we found no statistical evidence that the positions of the associated QTL were different (*Tip5* p = 0.55; *Tip8* p = 0.27; 400 bootstrap samples) (*Boehm et al., 2019*). These observations support the role of a single causal variant at each locus that is responsible for a pleiotropic trait. Coincident mapping can provide both additional statistical support for QTL (p values by Fisher's combined probability test: Chr 7, p = 0.067; Chr 2, p = 0.041) and suggests potential mechanisms of disease progression.

A number of factors can limit the statistical significance of QTL identified in the CC population, including small effect sizes, limited genotype availability, and the genetic complexity of the trait. We took an F$_2$ intercross approach to independently assess the importance of the lung CFU QTL on chromosomes 7 and 15 (*Tip8* and *Tip10*, *Table 1*). Given that the associations at both QTL were driven by the CAST haplotype (*Figure 2F*), we generated an F$_2$ population based on two CC strains, CC029/Unc and CC030/GeniUnc, that contained CAST sequence at *Tip8* and *Tip10*, respectively (*Figure 3A*

*and B*). The F$_2$ validation cohort (n = 251 mice) were genotyped (*Sigmon et al., 2020*) and infected with the *Mtb* strain H37Rv (IV route with infectious dose of 10$^5$ CFU, as per the original CC screen). At 1 month post infection, lung CFU was quantified, and we conducted QTL mapping in R/qtl2 (*Broman et al., 2019*) to identify host loci underlying bacterial burden in the lung. We identified a QTL significantly associated with lung CFU (LOD = 6.81; p < 0.05; 10,000 permutations) on chromosome seven that overlapped with *Tip8* (peak position Chr7:28.6 Mb), thus validating this locus as a main driver of bacterial burden. In this reduced complexity cross, we did not observe a QTL on chromosome 15 (*Tip10*). This may be due to the B6 haplotype at this locus in CC030, which did not represent the strongest phenotypic contrast to CAST. Additionally in the mapping validation study, we identified a new resistance (low lung CFU) locus on chromosome 8 (LOD = 4.08; peak position Ch8:116.1 Mb), driven by the CC029 cross partner with the CAST haplotype. This QTL was not present in the original CC screen, probably due to the low representation of the CAST haplotype at that marker in the CC cohort tested. Altogether, this intercross strategy validated *Tip8* as a strong predictor of lung CFU, though rigorous validation of *Tip10* may require a more optimal pairing of parental strains.

## *Mtb* adapts to diverse hosts by utilizing distinct gene repertoires

This survey of disease-associated traits demonstrated that the CC panel encompasses a number of qualitatively distinct immune phenotypes. To determine if different bacterial functions were necessary to adapt to these conditions, we leveraged transposon sequencing (TnSeq) as a high-resolution phenotyping approach to estimate the relative abundance of individual *Mtb* mutants after selection in each CC host genotype. To serve as benchmarks of known immunological lesions, we also performed TnSeq in B6 mice that were lacking the mediators of Th1 immunity, lymphocytes (*Rag2*$^{-/-}$) and IFNγ (*Ifng*$^{-/-}$), or were lacking the immunoregulatory mediators that control disease by inhibiting inflammation, nitric oxide synthase (*Nos2*$^{-/-}$) (*Mishra et al., 2013*) or the NADPH phagocyte oxidase (*Cybb*$^{-/-}$) (*Olive et al., 2018*). The relative representation of each *Mtb* mutant in the input library versus the output library recovered from each mouse spleen after one-month of infection was quantified by TnSeq (*Long et al., 2015*). A total of 123 saturated *Mtb* transposon libraries (representing >50,000 independent insertion events) were sequenced, capturing 60 distinct mouse genotypes (*Figure 4—source data 1*).

From this TnSeq screen, we identified 214 *Mtb* genes that are required for growth or survival of *Mtb* in B6 mice, based on significant underrepresentation of the corresponding mutant after four weeks of in vivo selection. Eighty-seven percent of these genes overlapped with a similar previous analysis in BALB/c mice (*Bellerose et al., 2020*) highlighting the specificity of the analysis. All but one of the genes found to be important in B6, were also required in the larger mouse panel, further increasing confidence in this *Mtb* gene set (*Figure 4A and B*). While the total number of genes found to be necessary in each genotype across the diversity panel was largely similar, the composition of these *Mtb* gene sets varied considerably. As more CC strains, and presumably more distinct immune states, were included in the analysis, the cumulative number of bacterial genes necessary for growth in these animals also increased. This cumulative gene set plateaued at ~750, after the inclusion of approximately 20–25 mouse genotypes (*Figure 4A*). Simply sampling additional libraries of B6 does not appreciably increase the number of genes identified as necessary for growth in that genotype (*Figure 4—figure supplement 1*), supporting the presence of alternative selective environments across the CC mice. The number of genes important for fitness in the CC panel far outnumbered the 380 genes identified in the B6 and immunodeficient KO strains combined (*Figure 4B* and *Figure 4—source data 1*).

To verify that our TnSeq study accurately assessed the effect of the corresponding loss-of-function alleles, we assessed the phenotypes of selected bacterial deletion mutants in a small set of mouse genotypes that were predicted to produce differential selection. Individual *Mtb* mutants lacking genes necessary for ESX-1 Type VII secretion (*eccB1*), siderophore-mediated iron acquisition (*mbtA*), phosphate transport (*pstC2*), glycerol catabolism (*glpK*), and RNA processing (*rnaseJ*) were generated and tagged with a unique molecular barcode. These mutants were combined with a barcoded wild-type parental strain, and the resulting 'mini-pool' was subjected to in vivo selection after IV infection of a sub-panel of mouse strains, as in the original screen. The relative abundance of each mutant was determined by sequencing the amplified barcodes and data from all reliably detected strains is shown in (*Figure 4*; *Figure 4—source data 2*). In each case, the difference in relative abundance predicted

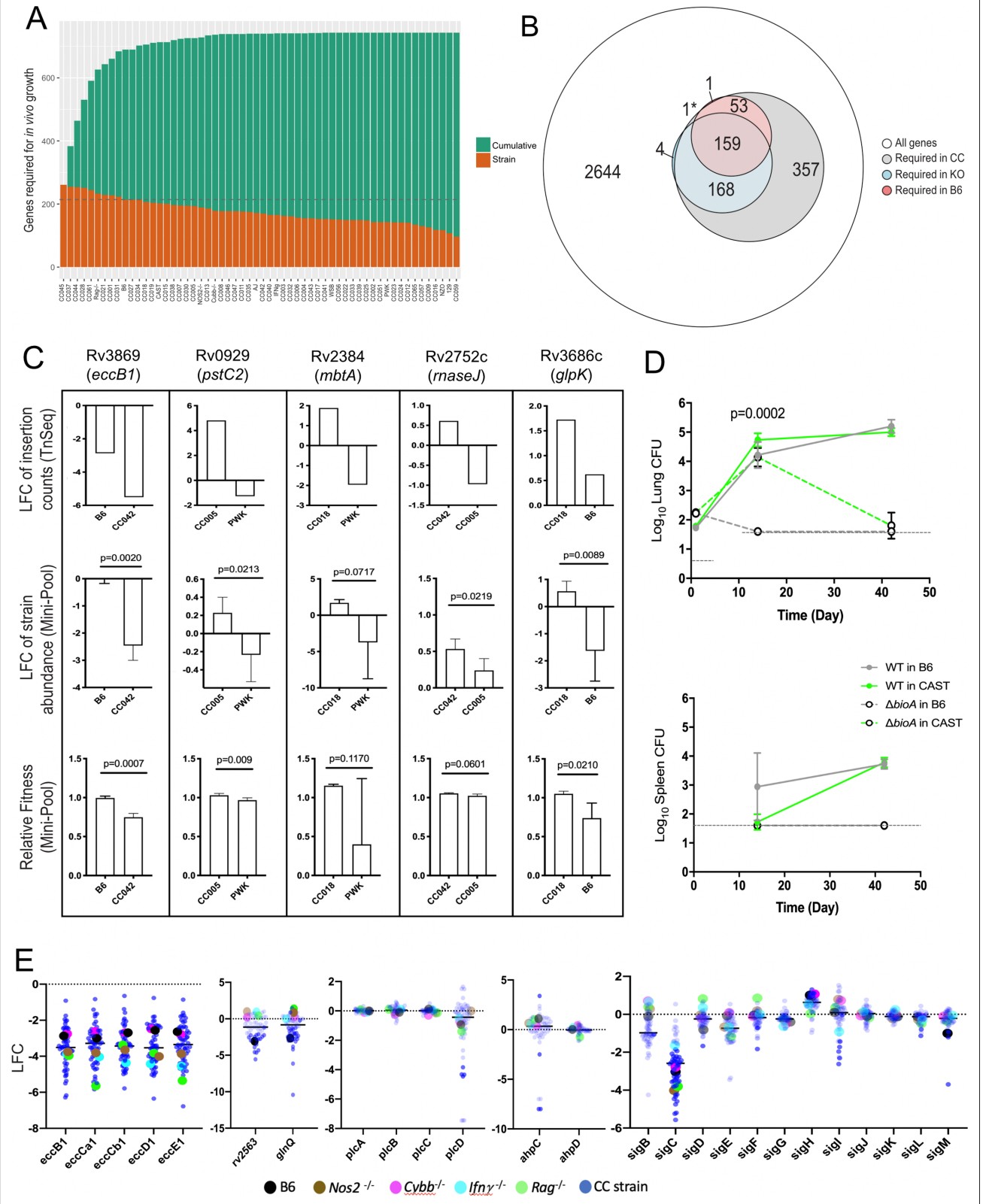

**Figure 4.** *Mtb* genetic requirements vary across diverse hosts. (**A**) The number of *Mtb* genes required for growth or survival in each diverse mouse strain across the panel (Qval ≤0.05). Orange indicates the mutants required for each strain; turquoise shows the cumulative requirement as each new host strain is added. (**B**) Venn diagram showing the composition of *Mtb* gene sets required in each category of host (white, largest circle), only required in the CC panel (gray), required in specific immunological KO mice (blue) and genes required in B6 mice (red). Note, 1* is required in B6 and KO. In order

*Figure 4 continued on next page*

*Figure 4 continued*

to be called 'essential' in each mouse strain, *Mtb* genes had to be significantly over or underrepresented in at least two genotypes. (**C**) Each box shows log$_2$ fold change (LFC) of individual mutants from the TnSeq screen relative to the input pool in indicated mouse strains (top); log$_2$ fold change of the indicated deletion mutants relative to WT from a pooled mutant validation infection (middle panel); relative fitness calculated from (middle panel) to account for generation differences in each host due to differential growth rate. Bars are the average of 3–6 mice per mutant/genotype ± SD. Statistical differences between mini-pool validation groups was assessed by Welch's t-test. (**D**) Lung CFU and spleen CFU from single strain low-dose aerosol infections of Δ*bioA* mutant or WT H37Rv strain in B6 and CAST mice at 2- or 5 weeks post-infection. Dashed line indicates the limit of detection. Each point indicates the average CFU ± SD of 4–5 mice per group. Statistical differences between groups were assessed by mixed effects models (Tukey's test). (**E**) Log$_2$ fold change of selected mutants from the TnSeq screen across the CC panel and immunological KO mice. Each dot represents the average LFC per mouse genotype; KO mouse strains (on a B6 background) dots are shown larger for clarity. All mice in the large CC TnSeq screen were male; mice in the Δ*bioA* aerosol validation were female; mice in the mini-pool validation studies were male and female with no significant differences detected. Source file of the TnSeq screen is available in *Figure 4—source data 1*; source count data of the TnSeq validation experiment is available in *Figure 4—source data 2*.

The online version of this article includes the following source data and figure supplement(s) for figure 4:

**Source data 1.** TnSeq summary table.

**Source data 2.** Validation counts table.

**Figure supplement 1.** Sampling additional B6 libraries does not appreciably increase the estimate of genes necessary for growth.

by TnSeq was reproduced with deletion mutants. In this simplified system, we were able to accurately quantify the expansion of the bacterial population and calculate the 'fitness' of each mutant relative to the wild-type strain. Fitness reflects the inferred doubling time of the mutant, where a fitness of 1 is defined as wild-type, and 0 represents a complete lack of growth. Even by this metric, the deletion mutants displayed the differences in fitness between mouse strains that was predicted by TnSeq (*Figure 4C*). The statistical significance of these differences in abundance or fitness were similar for each mutant (between p = 0.009 and p = 0.06), except for *mbtA* where the variation was higher, and confidence was modestly lower (p = 0.07 and p = 0.12). This study also allowed us to estimate the sensitivity of the TnSeq method, which could detect even the 30% fitness defect of the Δ*glpK* strain between the B6 and CC018 animals (*Figure 4C*), a defect that was not observed in previous studies in BALB/c mice (*Bellerose et al., 2019*; *Pethe et al., 2010*).

To also validate TnSeq predictions in a single-strain aerosol infection model, we used a biotin biosynthetic mutant. *bioA* is necessary for biotin production and is essential for growth in B6 mice (*Woong Park et al., 2011*). Our TnSeq study (*Figure 4—source data 1*) predicted this mutant was less attenuated in the CAST background (ratio of input/selected = 12.1) than in the B6 strain (ratio of input/selected = 42.2). Two weeks after aerosol infection, we found that the Δ*bioA* mutant was cleared from the lungs and spleen of B6 mice but displayed similar growth to wild-type in the lungs of CAST mice (*Figure 4D*). By 6 weeks post-infection the Δ*bioA* mutant had also been largely cleared from the lungs of CAST (*Figure 4D*). Thus, while TnSeq was unable predict long-term outcome, it provided an accurate assessment of relative growth attenuation in these host backgrounds.

## The immunological diversity of CC mice is reflected in the pathogen's genetic requirements

The distribution of *Mtb*'s requirements across the mouse panel suggested the presence of two broad categories of genes. A set of 136 'core' virulence functions were required in the majority of mouse genotypes, and a second larger set of 607 'adaptive' virulence genes were required in only a subset of lines (*Figure 4—source data 1*). The core functions included a number of genes previously found to be important in B6 mice, including those necessary for the synthesis of essential cofactors, such as pyridoxine (*pdx1*) (*Dick et al., 2010*); for the acquisition of nutrients, such as siderophore-bound iron (*irtAB*) (*Ryndak et al., 2010*), cholesterol (*mce4*) (*Pandey and Sassetti, 2008*), glutamine (*glnQ* and *rv2563*) (*Bellerose et al., 2020*) and for Type VII secretion (ESX1 genes) (*Stanley et al., 2003*). Despite the importance of these core functions, a large range in the relative abundance of these mutants was observed across the panel, and in some cases specific immunological requirements could be discerned. Mutants lacking the major structural components of the ESX1 system were attenuated for growth in B6 mice, as expected. This requirement was enhanced in mice lacking *Rag2*, *Ifng*, or *Nos2* (*Figure 4E*), consistent with the preferential role of ESX1 during the initial stage of infection before the initiation of adaptive immunity (*Stanley et al., 2003*), which is prolonged in these immunodeficient

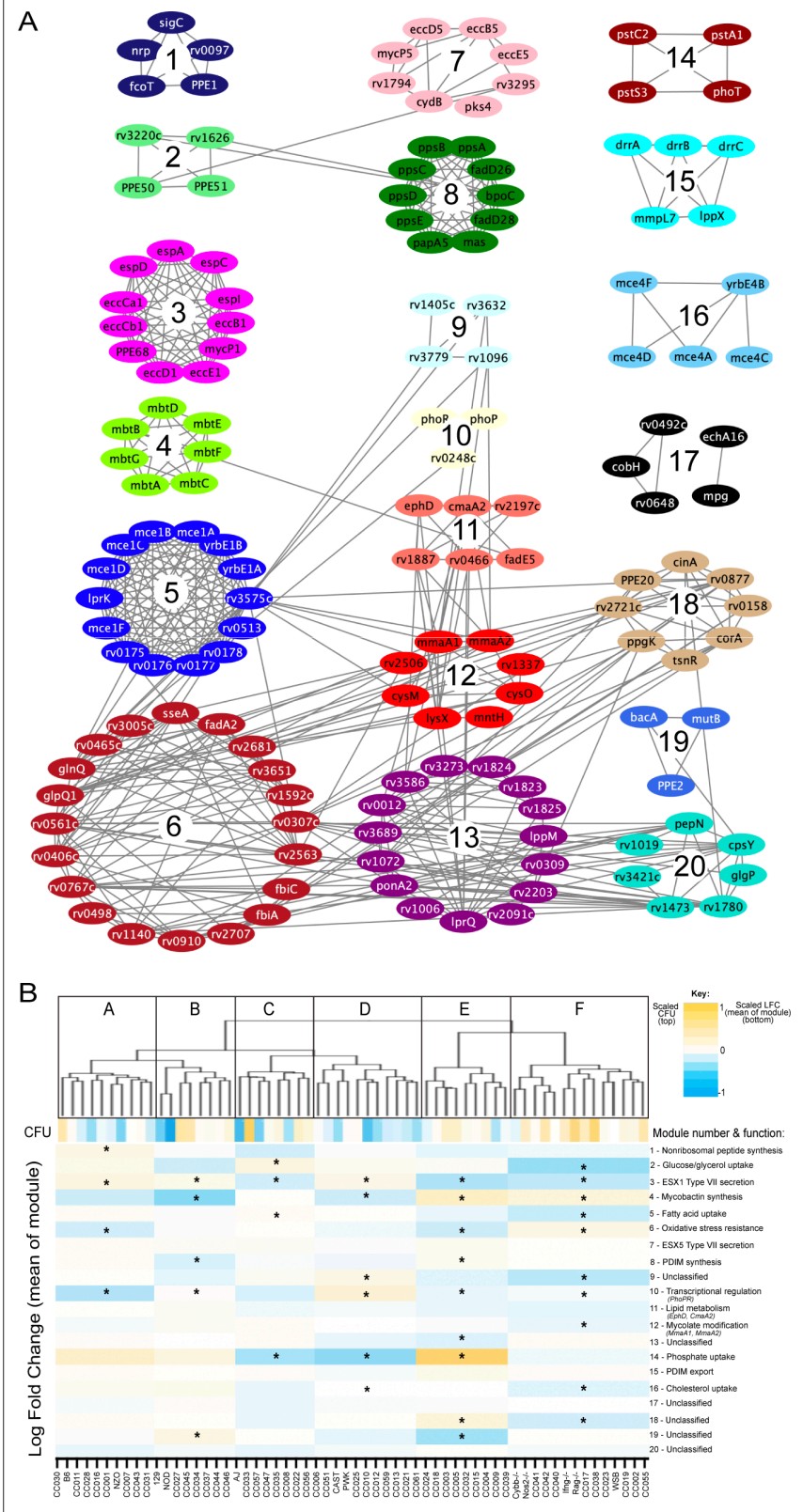

**Figure 5.** *Mtb* virulence pathways associate with distinct host immune pressures. (**A**) Weighted gene correlation network analysis (WGCNA) of the 679 *Mtb* genes that significantly vary across the diverse mouse panel. The most representative genes of each module (intramodular connectivity >0.6) are shown. (**B**) Mouse genotypes were clustered based on the relative abundance of the 679 variable *Mtb* mutants. The six major clusters (Cluster A-F)

*Figure 5 continued on next page*

*Figure 5 continued*

were associated with both CFU and the relative abundance of mutants in each bacterial module (1-20; right hand-side with known functions). Statistical analysis is described in Methods. Yellow shading indicates clusters associated with lung CFU. * indicate modules significantly associated with specific mouse clusters (p < 0.05).

The online version of this article includes the following figure supplement(s) for figure 5:

**Figure supplement 1.** Module-trait associations.

strains. In contrast, the attenuation of mutants lacking the *glnQ* encoded glutamine uptake system was relieved in all four immunodeficient mouse lines (*Figure 4E*). In both cases, the differential mutant abundance observed in these KO mice was reproduced, or exceeded, in the CC panel.

The adaptive virulence functions included a number of *Mtb* genes previously thought to be dispensable in the mouse model and were only necessary in CC strains. For example, the alkyl hydroperoxide reductase, AhpC has been proposed to function with the adjacently encoded peroxiredoxin, AhpD and is critical for detoxifying reactive nitrogen intermediates in vitro (*Chen et al., 1998*; *Hillas et al., 2000*). However, deletion of *ahpC* has no effect on *Mtb* replication in B6 or BALB/c mice (*Springer et al., 2001*), and we confirmed that *ahpC* and *ahpD* mutations had no effect in any of the B6-derived strains. In contrast, *ahpC*, but not *ahpD* mutants were highly attenuated in a small number of CC strains (*Figure 4E*). Similarly, the four phospholipase C enzymes of *Mtb* (*plcA-D*) are implicated in both fatty acid uptake and modifying host cell membranes but are dispensable for replication in B6 mice (*Le Chevalier et al., 2015*). Again, while we found that none of these genes were required in B6-derived KO mouse strains, the *plcD* mutants were specifically underrepresented in a number of CC mice (*Figure 4E*). These individual bacterial functions are controlled by regulatory proteins, such as the extracytoplasmic sigma factors. Despite the importance of these transcription factors in the response to stress, only *sigF* has consistently been shown to contribute to bacterial replication in standard inbred lines of mice (*Geiman et al., 2004*; *Rodrigue et al., 2006*). Our study assessed the importance of each sigma factor in parallel across diverse host genotypes and identified a clear role for several of these regulators. *sigC, sigI, sigF, sigL,* and *sigM* mutants were each significantly underrepresented in multiple strains of mice, and several of these phenotypes were only apparent in the diverse CC animals (*Figure 4E*). In sum, the 607 adaptive functions that are differentially required across the host panel represents nearly 20% of the non-essential gene set of *Mtb*, suggesting that a significant fraction of the pathogen's genome is dedicated to maintaining optimal fitness in diverse host environments.

## Differential genetic requirements define virulence pathways in *Mtb*

To more formally investigate the distinct stresses imposed on the bacterial population across this host panel, we characterized the differentially required bacterial pathways. Upon performing each possible pairwise comparison between the in vivo selected mutant pools, we found 679 mutants whose representation varied significantly (FDR < 5%) in at least two independent comparisons (*Figure 4—source data 1*). We then applied weighted gene correlation network analysis (WGCNA) (*Langfelder and Horvath, 2008*) to divide the mutants into 20 internally-correlated modules. Further enrichment of these modules for the most representative genes (intramodular connectivity >0.6) revealed that nearly all modules contained genes that are encoded adjacently in the genome and many of these modules consisted of genes dedicated to a single virulence-associated function (*Figure 5A*). Module three contains two distally encoded loci both known to be necessary for ESX1-mediated protein secretion, the primary ESX1 locus (*rv3868-rv3883*) and the *espACD* operon (*rv3616c-rv3614c*). Similarly, other modules consisted of genes responsible for ESX5 secretion (Module 7), mycobactin synthesis (Module 4), the Mce1 and Mce4 lipid importers (Modules 5 and 16), phthiocerol dimycocerosate synthesis (PDIM, Module 8), PDIM transport (Module 16), and phosphate uptake (Module 14). The 20 genes assigned to Module six included two components of an important oxidative stress resistance complex (*sseA* and *rv3005c*) and were highly enriched for mutants predicted to be involved in this same process via genetic interaction mapping (11/20 genes were identified in *Nambi et al., 2015*, a statistically significant overlap [p < 2.8e-10 by hypergeometric test]). Thus, each module represented a distinct biological function.

Many pathway-specific modules contained genes that represented novel functional associations. For example, the gene encoding the sigma factor, *sigC*, was found in Module one along with

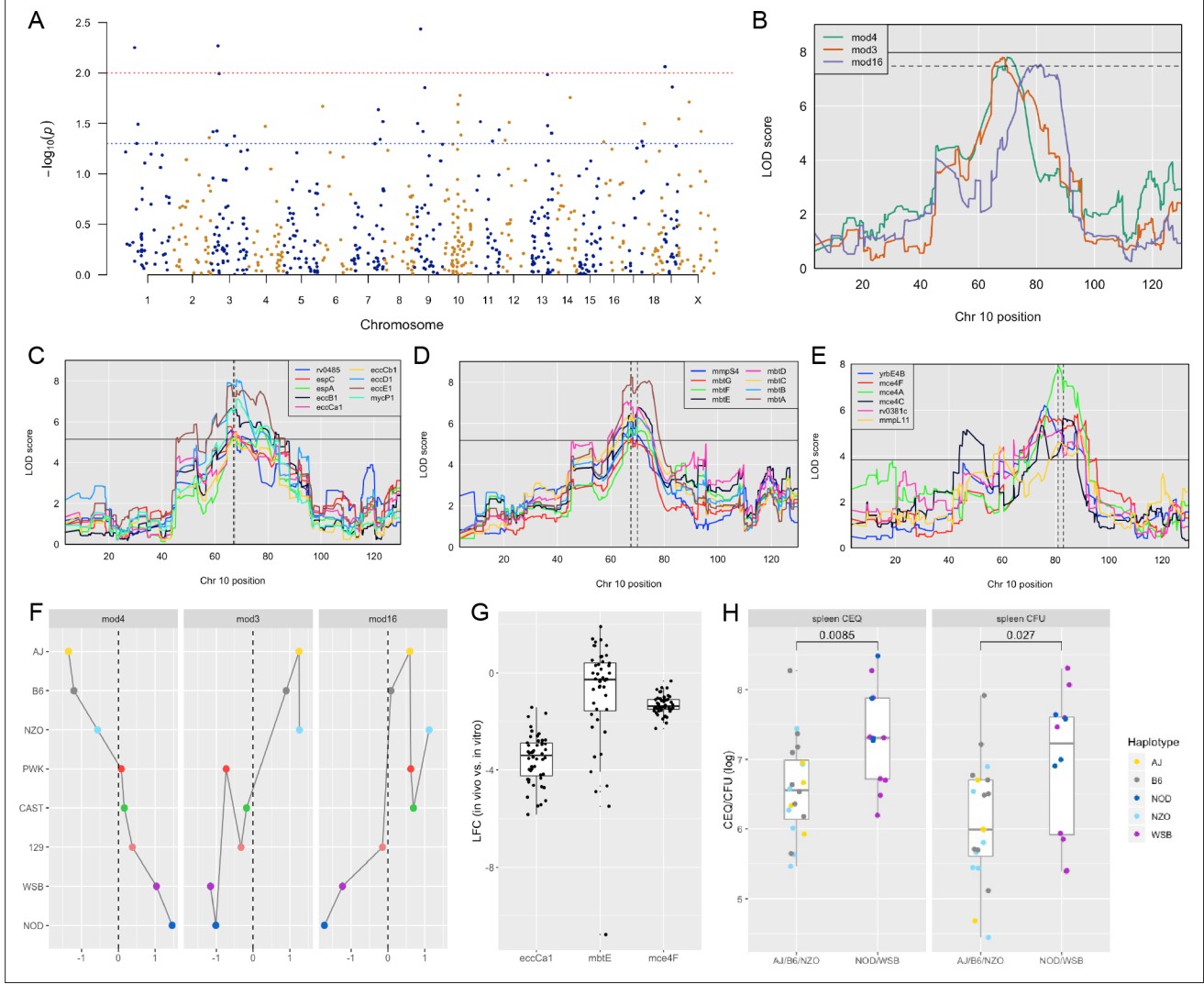

**Figure 6.** Identification of 'Host Interacting with Pathogen' QTL mapping (*HipQTL*). (**A**) Manhattan plot of single *Mtb* mutant QTL mapping across the mouse genome. Each dot represents an individual *Mtb* mutant plotted at the chromosomal location of its maximum LOD score. Red dashed line indicates p < 0.01; Blue p < 0.05. (**B**) Chromosome 10 QTL (in Mb) corresponding to *Mtb* eigentraits identified in network analysis in *Figure 5*. Module 3 (Type VII secretion, ESX1 operon; orange), Module 4 (Mycobactin synthesis, *mbt*; green) and Module 16 (Cholesterol uptake, *mce4*; purple) are shown. Solid and dotted lines indicated p = 0.05 and p = 0.1, respectively. Chromosomal position is in megabase units (Mb). (**C–E**) QTL mapping of single *Mtb* mutants corresponding to the (**C**) ESX1 module, (**D**) *mbt* module and (**E**) *mce4* modules. Coincidence of multiple QTL was assessed by the NL-method of *Neto et al., 2012*. Thresholds shown are for N = 9, N = 8, and N = 6 for panels C, D, and E, respectively. Chromosomal position is in megabase units (Mb). (**F**) Parental founder effects underlying Module 3, 4, and 16 QTL. Allele effects were calculated at the peak LOD score marker on chromosome 10. (**G**) Distribution of log₂ fold change (LFC) of representative single mutants from each module; *eccCa1* (ESX1 module), *mbtE* (*mbt* module), and *mce4F* (*mce4* module) relative to in vitro. Each dot is the LFC of the specified mutant in each CC mouse strain. Box and whiskers plots of each trait indicate the median and interquartile range. (**H**) Spleen CEQ and Spleen CFU for CC strains (box plots as in G). Mouse values are grouped by the parental haplotype allele series underlying the chromosome 10 *Hip42* locus (NOD/WSB vs AJ/B6/NZO). Each dot represents the average CFU/CEQ of each CC genotype. Statistical differences in disease-associated traits and distinct haplotypes groups were assessed by t-test. LOD, logarithm of the odds; LFC, log₂ fold change; CEQ, chromosomal equivalents; CFU, colony-forming units.

a non-ribosomal peptide synthetic operon. Previous genome-wide ChIP-seq and overexpression screens support a role for SigC in regulating this operon (*Minch et al., 2015*; *Turkarslan et al., 2015*). Similarly, *rv3220c* and *rv1626* have been proposed to comprise an unusual two-component system that is encoded in different regions of the genome (*Morth et al., 2005*). Both of these genes are found in Module 2, along with the PPE50 and PPE51 genes that encode at least one outer membrane channel (*Wang et al., 2020*; *Figure 5A*). In both cases, these associations support both regulatory and obligate functional relationships between these genes. Six of the 20 modules were not obviously

**Table 2.** *Hip*QTL for single *Mtb* mutant QTL and eigentrait/module QTL.
*Hip1-41* each represent host loci associated with the relative abundance of a single mutant (p < 0.05). *Hip42-46* correspond to *Mtb* eigentraits identified in network analysis in *Figure 5* (including significant p < 0.05 and suggestive p < 0.25). Figure column headings: QTL, quantitative trait loci; *Mtb, Mycobacterium tuberculosis*; Module #, module number determined from WGCNA modules; ORF, open reading frame; ID, identification number; LOD, logarithm of the odds; Chr, chromosome.

| QTL | Trait | *Mtb* ORF ID | Module # | LOD | P value | Chr | Start (Mb) | Peak (Mb) | End (Mb) |
|------|--------|--------------|----------|------|----------|-----|-----------|-----------|----------|
| *Hip1* | rv0770 | RVBD_0770 | mod17 | 9.81 | 5.61E-03 | 1 | 40.43 | 42.73 | 43.32 |
| *Hip2* | rv0309 | RVBD_0309 | mod13 | 7.95 | 3.22E-02 | 1 | 57.99 | 58.18 | 62.79 |
| *Hip3* | rv3657c | RVBD_3657 c | mod15 | 7.90 | 4.95E-02 | 1 | 136.39 | 138.24 | 143.60 |
| *Hip4* | rv0110 | RVBD_0110 | mod18 | 7.79 | 4.39E-02 | 2 | 170.67 | 174.00 | 178.84 |
| *Hip5* | rv3577 | RVBD_3577 | mod7 | 9.23 | 3.83E-02 | 3 | 3.32 | 10.03 | 14.67 |
| *Hip6* | rv3005c | RVBD_3005 c | mod6 | 8.03 | 3.75E-02 | 3 | 20.31 | 26.12 | 26.12 |
| *Hip7* | dinX | RVBD_1537 | mod15 | 9.97 | 5.38E-03 | 3 | 26.99 | 30.29 | 33.85 |
| *Hip8* | fadA6 | RVBD_3556 c | mod5 | 8.74 | 1.01E-02 | 3 | 29.23 | 35.22 | 37.11 |
| *Hip9* | dinX | RVBD_1537 | mod15 | 9.22 | 1.60E-02 | 3 | 36.22 | 36.83 | 38.27 |
| *Hip10* | rv2707 | RVBD_2707 | mod6 | 8.17 | 4.21E-02 | 3 | 100.90 | 103.23 | 115.82 |
| *Hip11* | rv3701c | RVBD_3701 c | mod6 | 7.90 | 3.38E-02 | 4 | 74.00 | 78.25 | 87.00 |
| *Hip12* | ahpC | RVBD_2428 | mod13 | 8.12 | 2.14E-02 | 6 | 19.75 | 22.21 | 23.31 |
| *Hip13* | umaA | RVBD_0469 | mod20 | 8.32 | 2.31E-02 | 7 | 117.87 | 118.41 | 120.15 |
| *Hip14* | rv2566 | RVBD_2566 | mod15 | 7.86 | 4.55E-02 | 7 | 123.21 | 126.67 | 126.67 |
| *Hip15* | rv3173c | RVBD_3173 c | mod5 | 8.17 | 3.03E-02 | 7 | 137.41 | 138.36 | 138.36 |
| *Hip16* | rv3173c | RVBD_3173 c | mod5 | 8.12 | 3.28E-02 | 7 | 139.15 | 140.76 | 141.88 |
| *Hip17* | rv3502c | RVBD_3502 c | mod5 | 8.16 | 3.17E-02 | 9 | 15.91 | 16.33 | 18.72 |
| *Hip18* | mycP1 | RVBD_3883 c | mod3 | 9.09 | 3.66E-03 | 9 | 28.47 | 29.45 | 31.10 |
| *Hip19* | rv0057 | RVBD_0057 | mod6 | 8.39 | 3.79E-02 | 9 | 36.78 | 40.07 | 40.36 |
| *Hip20* | hycE | RVBD_0087 | mod20 | 8.21 | 1.40E-02 | 9 | 47.40 | 47.93 | 51.80 |
| *Hip21* | mbtA | RVBD_2384 | mod4 | 8.30 | 2.05E-02 | 10 | 64.48 | 68.09 | 75.42 |
| *Hip22* | eccD1 | RVBD_3877 | mod3 | 8.08 | 3.08E-02 | 10 | 64.56 | 68.12 | 71.04 |
| *Hip23* | rv2989 | RVBD_2989 | mod12 | 9.16 | 1.67E-02 | 10 | 74.30 | 77.63 | 81.03 |
| *Hip24* | mce4A | RVBD_3499 c | mod16 | 7.91 | 4.12E-02 | 10 | 78.88 | 81.36 | 88.25 |
| *Hip25* | treS | RVBD_0126 | mod7 | 7.94 | 3.04E-02 | 11 | 20.80 | 36.14 | 44.06 |
| *Hip26* | pckA | RVBD_0211 | mod3 | 7.67 | 4.74E-02 | 11 | 85.95 | 89.78 | 91.75 |
| *Hip27* | aspB | RVBD_3565 | mod7 | 8.32 | 3.66E-02 | 11 | 114.69 | 116.99 | 117.08 |
| *Hip28* | rv1227c | RVBD_1227 c | mod17 | 9.16 | 4.64E-02 | 12 | 25.23 | 25.23 | 28.54 |
| *Hip29* | rv0219 | RVBD_0219 | mod20 | 7.94 | 3.09E-02 | 12 | 40.65 | 42.65 | 47.22 |
| *Hip30* | rv3643 | RVBD_3643 | mod8 | 8.89 | 1.04E-02 | 13 | 95.43 | 97.08 | 97.79 |
| *Hip31* | ansA | RVBD_1538 c | mod11 | 8.28 | 3.32E-02 | 13 | 96.82 | 97.79 | 99.09 |
| *Hip32* | echA19 | RVBD_3516 | mod20 | 9.68 | 3.95E-02 | 13 | 113.20 | 114.59 | 117.64 |
| *Hip33* | rv1836c | RVBD_1836 c | mod15 | 9.19 | 1.75E-02 | 14 | 74.94 | 76.40 | 76.43 |
| *Hip34* | rv2183c | RVBD_2183 c | mod11 | 7.75 | 4.84E-02 | 16 | 12.18 | 14.06 | 17.92 |

*Table 2 continued on next page*

*Table 2 continued*

| QTL | Trait | *Mtb* ORF ID | Module # | LOD | P value | Chr | Start (Mb) | Peak (Mb) | End (Mb) |
|---|---|---|---|---|---|---|---|---|---|
| *Hip35* | rv1178 | RVBD_1178 | mod6 | 8.19 | 4.76E-02 | 17 | 80.92 | 80.92 | 83.23 |
| *Hip36* | rv0492c | RVBD_0492 c | mod17 | 8.90 | 3.18E-02 | 18 | 5.85 | 5.85 | 12.40 |
| *Hip37* | cysM | RVBD_1336 | mod12 | 8.47 | 8.67E-03 | 19 | 4.20 | 6.46 | 6.46 |
| *Hip38* | atsA | RVBD_0711 | mod1 | 8.58 | 1.38E-02 | 19 | 31.21 | 37.86 | 37.93 |
| *Hip39* | galE2 | RVBD_0501 | mod6 | 8.10 | 2.85E-02 | X | 6.01 | 6.01 | 9.12 |
| *Hip40* | pks11 | RVBD_1665 | mod17 | 8.25 | 1.94E-02 | X | 50.43 | 51.75 | 52.29 |
| *Hip41* | pknK | RVBD_3080 c | mod17 | 8.73 | 3.79E-02 | X | 95.01 | 102.02 | 130.04 |
| *Hip42* | Module 3 | ESX1 operon | mod3 | 7.80 | 5.38E-02 | 10 | 64.7 | 68.27 | 77.07 |
| *Hip42* | Module 4 | Mycobactin (*mbt*) | mod4 | 7.79 | 5.05E-02 | 10 | 65.23 | 69.94 | 74.30 |
| *Hip43* | Module 16 | *mce4* operon | mod16 | 7.53 | 7.97E-02 | 10 | 74.30 | 81.36 | 87.61 |
| *Hip44* | Module 19 | unclassified | Module 19 | 7.64 | 1.39E-01 | 11 | 60.87 | 62.20 | 63.26 |
| *Hip45* | Module 10 | Transcriptional regulation | Module 10 | 6.95 | 1.04E-01 | 15 | 100.39 | 102.25 | 103.36 |
| *Hip46* | Module 10 | Transcriptional regulation | Module 10 | 6.32 | 2.54E-01 | 19 | 32.74 | 32.87 | 37.48 |

enriched for genes of a known pathway, demonstrating that novel virulence pathways are important for adapting to changing host environments.

To explore the complexity of immune environments in the CC, we used the TnSeq profiles of the 679 differentially fit *Mtb* mutants to cluster the mouse panel into six major groups of host genotypes (*Figure 5B*). One mouse cluster was significantly associated with high CFU (Cluster F, *Figure 5B*), which contained susceptible *Nos2*[-/-], *Cybb*[-/-], *Ifng* [-/-], and *Rag2*[-/-] animals. This high CFU cluster was associated with alterations in a diverse set of bacterial modules and corresponded to an increased requirement for lipid uptake (Modules 5 and 16) and ESX1, consistent with previous TnSeq studies in susceptible *Nos2*[-/-] and C3HeB/FeJ mice (*Mishra et al., 2017*). In addition, we identified a significant reduction in the requirement for the oxidative stress resistance (Module 6) in the highest CFU cluster. Despite these associations between bacterial genetic requirements and susceptibility, the clustering of mouse genotypes was largely independent of overall susceptibility. Similarly, while Module one was significantly associated with high IFN$\gamma$ levels, other bacterial fitness traits were not highly correlated with cytokine abundance (*Figure 5—figure supplement 1*). Instead, each major mouse cluster was associated with a distinct profile of *Mtb* genetic requirements. This observation supported the presence of qualitatively distinct disease states and complex genetic control of immunity.

## Identification of genome-wide host interacting with pathogen QTL (*Hip*QTL)

To investigate the host genetic determinants of the bacterial microenvironment, we leveraged TnSeq as a high-resolution phenotyping platform to associate *Mtb* mutant fitness profiles with variants in the mouse genome. When the relative abundance of each *Mtb* mutant phenotype was considered individually, the corresponding 'Host Interacting with Pathogen QTL' (*Hip*QTL) were distributed across the mouse genome (*Figure 6A*). Forty-one of these traits reached an unadjusted p-value threshold of 0.05 and can be considered as robust for single hypothesis testing (*Hip1-41*, *Table 2*). These included *Hip*QTL associated with *ahpC* (*Hip12*) and *eccD1* (*Hip22*), that explain at least a portion of the observed variable abundance of these *Mtb* mutants (*Figure 4E*). In order to reduce complexity and increase the power of this analysis, we performed QTL mapping based on the first principal component of each of the previously defined modules of *Mtb* virulence pathways (*Figure 5A*). Three of these 'eigentraits' were associated with QTL at a similar position on chromosome 10 (*Figure 6B*), corresponding to

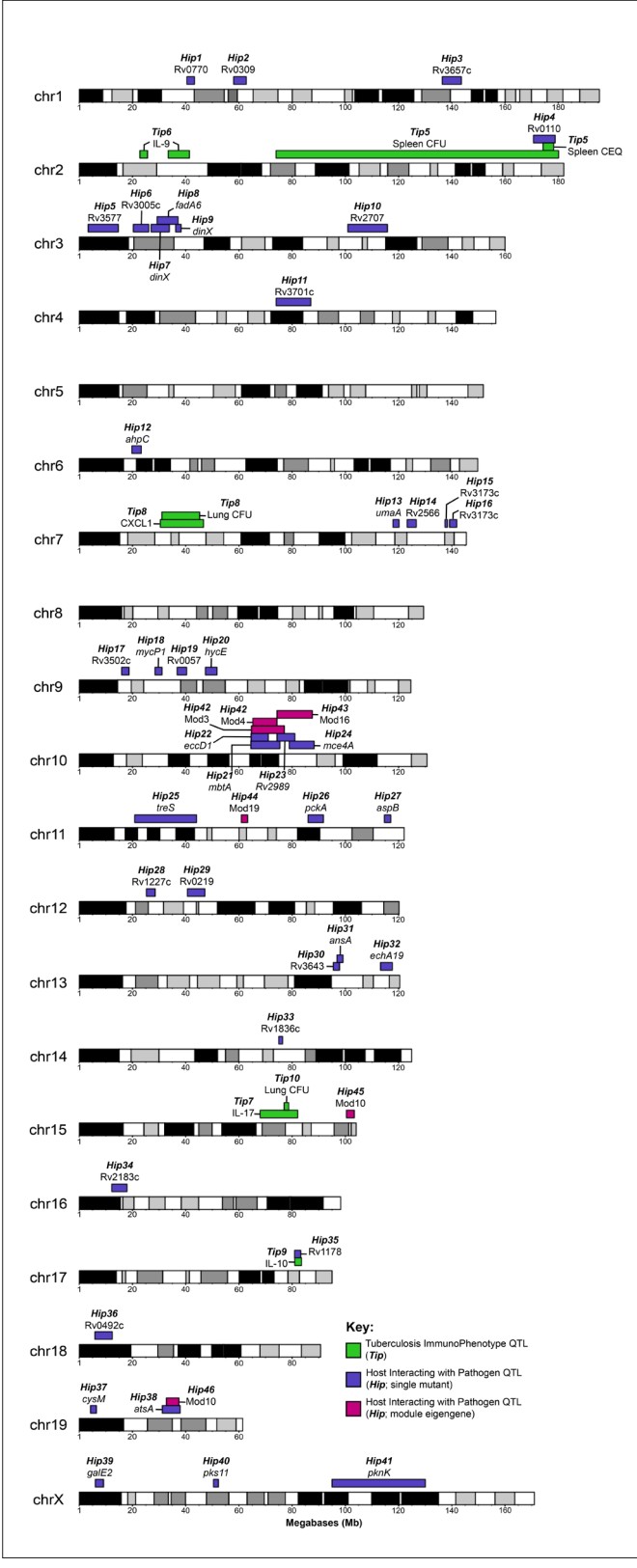

**Figure 7.** Visual representation of all *Tip* and *Hip*QTL mapped in the CC TnSeq infection screen. Tuberculosis ImmunoPhenotypes (*Tip*) QTL (QTL mapped by disease-associated traits in CC mice), are shown in green. *Tip*QTL mapped by separate traits that share similar founder effects were considered to be the same QTL and were named accordingly. Host Interacting with Pathogen (*Hip*) QTL, (QTL mapped by individual TnSeq mutant relative

*Figure 7 continued on next page*

*Figure 7 continued*

abundance profiles), are shown in purple. After WGCNA mutant clustering and mapping with representative eigengenes from each module, QTL mapped by module eigengenes are shown in magenta.

Module 3 (TypeVII secretion, ESX1), Module 4 (mycobactin synthesis, *mbt*), or Module 16 (cholesterol uptake, *mce4*). In all three cases, a single mutant from the module was independently associated with a QTL at the same position as the module eigentrait (*Table 2*; *Hip21, Hip22, Hip24*), and all genes in the corresponding network cluster (*Figure 5A*) mapped to the same location (*Figure 6C–E*). While not all individual traits mapped with high confidence, the coincidence of these multiple QTL was statistically significant (*Figure 6B*).

Both the relative positions of the module-associated QTL and the associated founder haplotypes indicated that a single genetic variant controlled the abundance of ESX1 and *mbt* mutants (*Hip42*). Specifically, we found no statistical support for differentiating these QTL based on position (p = 0.93) (*Boehm et al., 2019*) and the same founder haplotypes were associated with extreme trait values at both loci, though they had opposite effects on the abundance of ESX1 mutants and *mbt* mutants (*Figure 6F*). We conclude that a single haplotype has a pleiotropic effect on *Mtb*'s environment and has opposing effects of the requirement for mycobactin synthesis and ESX1 secretion. The relationship between this variant and the *mce4*-associated QTL (*Hip43*) was less clear, as the statistical support for independent QTL was weak (ESX1 and *mce4* QTL p = 0.17; *mbt* and *mce4* QTL p = 0.08) and the effects of founder haplotypes were similar but not identical (*Figure 6F*). Some of this ambiguity may be related to the relatively small range in trait values for *mce4*, compared to either ESX1 or *mbt* (*Figure 6G*). Based on this data, we report two distinct *Hip*QTL in this region (*Hip42* and *43*; *Table 2*).

Two *Tip*QTL overlapped with *Hip*QTL (*Figure 7*; *Tip5/Hip4* on chromosome two and *Tip9/Hip35* on chromosome 17), suggesting specific interactions between bacterial fitness and immunity. However, most *Tip*- and *Hip*QTL were distinct, indicating that the fitness of sensitized bacterial mutants can be used to detect genetic variants that subtly influence the bacterial environment but not overtly alter disease. We chose to further investigate whether *Hip*QTL might alter overall bacterial disease using the most significant *Hip*QTL on chromosome 10 (*Hip42*). We found that the founder haplotypes associated with extreme trait values at this QTL could differentiate CC strains with significantly altered total bacterial burden, and the NOD and WSB haplotypes were associated with higher bacterial numbers (p = 0.0085 for spleen CEQ; P = 0.027 for spleen CFU; *Figure 6H*). Thus, not only could the *Hip*QTL strategy identify specific interactions between host and bacterial genetic variants, but it also appears to be a sensitive approach for identifying host loci that influence the trajectory of disease.

## Identifying candidate genes underlying QTL

A pipeline was designed to prioritize genetic variants based on genomic and tuberculosis disease criteria. We concentrated on three QTL: two that were highly significant and with clear allele effects (*Tip5, Hip42*), and the *Tip8* locus which we validated by intercross. For each QTL region, we identified genes that belonged to a differentially expressed transcriptional module in mouse lungs following *Mtb* infection (*Moreira-Teixeira et al., 2020*). Next, we identified genetic variants segregating between the causal CC haplotypes in the gene bodies corresponding to these transcripts, and prioritized missense or nonsense variants.

For the *Tip5* QTL underlying CEQ, CFU, and IL-10 levels, we identified nine candidate genes with regulatory or splicing variants and two genes with missense variants specific to the NOD haplotype. Of these candidates, cathepsin Z (*Ctsz*) encodes a lysosomal cysteine proteinase and has previously been associated with TB disease risk in humans (*Adams et al., 2011*; *Cooke et al., 2008*). The QTL underlying lung CFU and CXCL1 abundance (*Tip8*), which was driven solely by the genetically divergent CAST founder haplotype, contained over 50 genes (*Table 3*) and will need further refinement. The QTL associated with the abundance of ESX1 and *mbt* mutants (*Hip42*) had a complex causal haplotype pattern (AJ/B6/NZO vs. 129/CAST/PWK vs. NOD/WSB) suggesting multiple variants might be impacting common genes. Within this interval, we identified 13 genes expressed in response to *Mtb* infection, three of which had SNPs fully or partially consistent with at least one of the identified causal haplotype groups (*Table 3*). *Ank3* contains several SNPs in the 3' UTR and other non-coding exons that differentiated NOD/WSB from the other haplotypes. Similarly, *Fam13c* had two missense mutations following the same haplotype pattern. For the AJ/B6 haplotype state, we identified a

**Table 3.** Candidate genes within QTL regions. Prioritized candidates shown for selected QTL. Candidates were prioritized by filtering on (1) differential expression during *Mtb* infection, and (2) variants within TB-expressed genes that segregated between informative CC haplotypes. Genes listed below contain non-synonymous variants (i.e. amino acid changes, regulatory mutations or splicing mutations) consistent with the identified singly causal haplotype (NOD for *Tip5*; CAST for *Tip8*). *Hip42* displayed a more complex haplotype pattern (WSB/NOD vs AJ/B6/NZO), and candidate selection is discussed in the main text. Genes with missense or nonsense variants (denoted by *).

| Tip5 | Tip8 | | Hip42 |
|---|---|---|---|
| Ctsz | Fxyd5* | Siglecg | Ank3 |
| Tubb1 | Fxyd1 | Nkg7 | Cdk1 |
| Atp5e | Lgi4 | Cd33* | Tmem26 |
| Prelid3b | Fxyd3 | Siglece* | Slc16a9 |
| Zfp831* | Hpn | Klk13 | Fam13c |
| Edn3 | Scn1b | Klk8 | Rhobtb1 |
| Gm14391* | Gramd1a* | Klk7* | |
| Gm6710 | Pdcd2l* | Klk1b9* | |
| Zfp931 | Gpi1 | Klk1 | |
| | 4931406P16Rik | Clec11a | |
| | Kctd15 | Shank1 | |
| | Chst8 | Syt3 | |
| | Pepd | Lrrc4b | |
| | Cebpa | Josd2 | |
| | Slc7a10 | Spib | |
| | Lrp3* | Pold1 | |
| | Rhpn2 | Napsa* | |
| | Faap24 | Kcnc3 | |
| | Tdrd12* | Myh14 | |
| | Ankrd27* | Atf5 | |
| | Pdcd5 | Il4i1 | |
| | Dpy19l3 | Pnkp* | |
| | Tshz3* | Ptov1 | |
| | Ccne1 | Fuz | |
| | 1600014C10Rik | Tsks | |
| | Plekhf1 | Cpt1c* | |
| | Vstm2b | | |
| | Zfp975* | | |
| | Zfp715* | | |

missense mutation and several variants in the 3' UTR of *Rhobtb1*, which belongs to the Rho family of the Ras superfamily of small GTPases (**Goitre et al., 2014**). Overall, the evidence supports a role for *Rhobtb1* in a monogenic effect at the chromosome 10 locus. This evidence includes both protein coding differences dividing AJ/B6 from the other haplotypes, a potential expression/transcript regulatory difference that segregates the NOD/WSB state from the remaining parental haplotypes, and a plausible role for this gene in controlling intracellular trafficking (**Long et al., 2020**) and the opposing requirements for ESX1 and mycobactin.

## Discussion

Our broad profiling of both host and pathogen traits after *Mtb* infection in a large panel of CC strains, created a reproducible resource to study the diverse host-pathogen interactions that drive tuberculosis disease. The immunological analysis of the CC panel identified correlates of TB disease progression that were consistent with previous studies in both mice and human patients (**Ahmed et al., 2020**; **Niazi et al., 2015**; **Zak et al., 2016**). We also identified outlier strains that produce distinct immunological states, suggesting that our previous reliance on genetically homogenous lab strains of mice has oversimplified our understanding of TB pathogenesis. For example, despite the strong correlation between lung bacterial burden and weight loss, CC030/GeniUnc and CC040/TauUnc mice suffered from more inflammation and wasting than would be predicted from the number of bacteria in their lungs or spleens. This phenotype reflects a failure of disease 'tolerance', which is proposed to be a critical determinant of protective immunity (**Ayres and Schneider, 2012**; **Olive et al., 2018**). Similarly, we identified a number of CC genotypes that produce very low, or undetectable, levels of the protective cytokine IFNγ, but still control lung bacterial replication. While a growing body of literature suggests that immune responses distinct from the canonical Th1 response can control infection (**Lu et al., 2019**; **Sakai et al., 2016**), these CC strains are the first example of an animal model in which IFNγ appears to be dispensable. Despite the relatively small group sizes used in this initial phenotypic screen, the reproducibility of the CC strains facilitated the identification of these phenotypes and provides tractable models for further characterization.

The ability to separate aspects of the immune response from disease progression implied

that these features are under distinct multigenic control. Our study demonstrated the feasibility of mapping the genetic variants that control the complex immune response to *Mtb*. The QTL identified in this study are generally distinct from CC loci that control immunity to viruses (*Ferris et al., 2013*; *Gralinski et al., 2017*; *Noll et al., 2020*) or another intracellular bacterial pathogen, *Salmonella* (*Zhang et al., 2019*). However, *Tip8* and *Tip10* overlap with QTL previously defined via *Mtb* infection of a CC001xCC042 F$_2$ intercross population (*Smith et al., 2019*) suggesting that common variants may have been identified in both studies. While the specific genetic variants responsible for these QTL remain unknown, both coincident trait mapping and bioinformatic analysis suggest mechanistic explanations for some QTL-phenotype associations. For example, a single interval on chromosome two controls CFU levels and IL-10, and contains a variant in the *Ctsz* gene encoding Cathepsin Z. *Ctsz* is a strong candidate considering its known roles in autophagy (*Amaral et al., 2018*), dendritic cell differentiation and function (*Obermajer et al., 2008*), its upregulation in non-human primates (*Ahmed et al., 2020*) and human patients with *Mtb* (*Zak et al., 2016*), and the association of *CTSZ* variants with disease risk in human TB studies (*Adams et al., 2011*; *Cooke et al., 2008*). Regardless of the responsible variants, these data will facilitate the generation of new congenic animal models that isolate the contribution of each QTL to phenotype.

Using TnSeq as a multidimensional phenotyping method across this population provided insight into how the diversity of host-derived microenvironments have shaped the pathogen's genome. While *Mtb* is an obligate pathogen with no significant environmental niche, only a minority of the genes in its genome have been found to contribute to bacterial fitness in either laboratory media or individual inbred mouse models, leaving the pressures that maintain the remaining genomic content unclear. Our study indicated that a roughly similar number of genes are important for *Mtb* fitness in a given mouse strain, even immunodeficient strains that likely represent the most divergent environments. While this observation may seem counterintuitive, it is consistent with previous TnSeq studies in both mouse models (*Mishra et al., 2017*) and in vitro conditions (*Minato et al., 2019*), where distinct but similarly sized gene sets are necessary for growth under very different conditions. Overall, we find that approximately three times more genes contribute to bacterial growth or survival in the CC population than in the standard B6 model. While some bacterial genetic requirements could be associated with known immune pathways, most of the differential pressures on bacterial mutants could not be attributed to these simple deficiencies in known mechanisms of immune control. Instead, it appears that the CC population produces a spectrum of novel environments, and that a relatively large fraction of the pathogen's genome is needed to adapt to changing immune pressures. Differential pressures on these adaptive virulence functions are similarly apparent in genomic analyses of *Mtb* clinical isolates. Signatures of selection have been detected in ESX1-related genes (*Holt et al., 2018*; *Sousa et al., 2020*), *phoPR* (*Gonzalo-Asensio et al., 2014*), and the oxidative stress resistance gene *sseA* (*de Keijzer et al., 2014*), suggesting that *Mtb* is exposed to similarly variable host pressures in genetically diverse human and mouse populations. While the combinatorial complexity of associating host and pathogen genetic variants in natural populations is daunting, the identification of *Hip*QTL in the CC panel indicates that these inter-species genetic interactions can be important determinants of pathogenesis and can be dissected using this tractable model of diversity.

# Materials and methods

**Key resources table**

| Reagent type (species) or resource | Designation | Source or reference | Identifiers | Additional information |
|---|---|---|---|---|
| Strain, strain background (*Mus musculus*, male) | Collaborative Cross mice | DOI: https://doi.org/10.1038/ng1104-113 | | |
| Strain, strain background (*Mycobacterium tuberculosis*) | H37Rv | DOI: 10.1073/pnas.2134250100 | | |
| Genetic reagent (*Mycobacterium tuberculosis*) | ΔglpK; ΔpstC2; ΔeccB1; ΔmbtA; | DOI: 10.1128/mBio.01467–18 | | |
| Genetic reagent (*Mycobacterium tuberculosis*) | ΔBioA | DOI: 10.1371/journal.ppat.1002264 | | |

*Continued on next page*

Continued

| Reagent type (species) or resource | Designation | Source or reference | Identifiers | Additional information |
|---|---|---|---|---|
| Recombinant DNA reagent | pKM464 (plasmid) | DOI: 10.1128/mBio.01467–18 | | |
| Recombinant DNA reagent | Barcode qtag (plasmid) | DOI: 10.1128/mSystems.00396–20 | | |
| Sequence-based reagent | qtag/barcode sequencing primer sets | DOI: 10.1128/mSystems.00396–20 | Table S6 | |
| Sequence-based reagent | MiniMUGA genotyping array | Neogen Inc | | |
| Sequence-based reagent | GigaMUGA genotyping array | Neogen Inc | | |
| Commercial assay or kit | 32-plex cytokine assay | Eve Technologies, Calgary, CA | | |
| Software, algorithm | R/qtl2 | DOI: 10.1534/genetics.118.301595 | | Dr. Karl Broman (University of Wisconsin-Madison) |
| Software, algorithm | WGCNA | DOI: 10.1186/1471-2105-9-559 | | Dr. Peter Langfelder (UCLA) |

## Mice

Male and female Collaborative Cross parental strains (A/J #0646; C57BL/6 J #0664; 129S1/SvImJ #02448; NOD/ShiLtJ #01976; NZO/HiLtJ #02105; CAST/EiJ #0928, PWK/PhJ #3,715 and WSB/EiJ #01145) and single gene immunological knockout mice were purchased from The Jackson Laboratory ($Nos2^{-/-}$ #2609, $Cybb^{-/-}$ #2365, $Ifn\gamma^{-/-}$ #2287) or Taconic ($Rag$N12) and bred at UMASS. Male mice from 52 CC strains were purchased from the UNC Systems Genetics Core Facility (SGCF) between July 2013 and August 2014. The 52 CC strains used in this study include: CC001/Unc, CC002/Unc, CC003/Unc, CC004/TauUnc, CC005/TauUnc, CC006/TauUnc, CC007/Unc, CC008/GeniUnc, CC009/Unc, CC010/GeniUnc, CC011/Unc, CC012/GeniUnc, CC013/GeniUnc, CC015/Unc, CC016/GeniUnc, CC017/Unc, CC018/Unc, CC019/TauUnc, CC021/Unc, CC022/GeniUnc, CC023/GeniUnc, CC024/GeniUnc, CC025/GeniUnc, CC027/GeniUnc, CC028/GeniUnc, CC029/Unc, CC030/GeniUnc, CC031/GeniUnc, CC032/GeniUnc, CC033/GeniUnc, CC034/Unc, CC035/Unc, CC037/TauUnc, CC038/GeniUnc, CC039/Unc, CC040/TauUnc, CC041/TauUnc, CC042/GeniUnc, CC043/GeniUnc, CC044/Unc, CC045/GeniUnc, CC046/Unc, CC047/Unc, CC051/TauUnc, CC055/TauUnc, CC056/GeniUnc, CC057/Unc, CC059/TauUnc, CC060/Unc, CC061/GeniUnc, CC065/Unc, CC068/TauUnc. More information regarding the CC strains can be found at http://csbio.unc.edu/CCstatus/index.py?run=AvailableLines.information.

CC030 x CC029 $F_2$ mice were generated in the FPMV lab at UNC by crossing CC030 and CC029 mice (purchased from the SGCF in 2016) to generate $F_1$s (both CC030 dam by CC029 sires as well as CC029 dam by CC030 sires). The resulting $F_1$s were subsequently intercrossed to generate 251 $F_2$ mice with all possible grandparental combinations. Male and female $F_2$ mice were shipped to UMASS for $Mtb$ infections.

All mice were housed in a specific pathogen-free facility under standard conditions (12 hr light/dark, food and water ad libitum). Mice were infected with $Mtb$ between 8 and 12 weeks of age. Male mice were used for initial large CC screen; male and female mice were used for F2 validation cohort.

## *M. tuberculosis* trains

All *M. tuberculosis* strains (H37Rv background) were grown in Middlebrook 7H9 medium containing oleic acid-albumin-dextrose-catalase (OADC), 0.2% glycerol, and 0.05% Tween 80 to log-phase with shaking (200 rpm) at 37 °C. Hygromycin (50 μg/ml) or kanamycin (20 μg/ml) was added when necessary. The TnSeq library consisting of *Himar1* transposon mutants was described previously (*Sassetti et al., 2003*). The ΔbioA strain was made by homologous recombination as previously described (*Woong Park et al., 2011*). For pooled mutant infections, deletion strains (GlpK, PstC2, EccB1, MbtA) were constructed using ORBIT (*Murphy et al., 2018*), which included gene replacement by the vector pKM464 carrying unique q-Tag sequences to identify each mutant for deep sequencing. The *rnaseJ*

mutant was also made by ORBIT and was kindly provided by Dr. Nathan Hicks and Dr Sarah Fortune. Prior to all in vivo infections, cultures were washed, resuspended in phosphate-buffered saline (PBS) containing 0.05% Tween 80, and sonicated before diluting to desired concentration (see below).

## Mouse infections

For TnSeq experiments, $1 \times 10^6$ CFU of a saturated library of *Himar1* transposon mutants (*Sassetti et al., 2003*) was delivered via intravenous tail vein injection, resulting in an infectious dose (Day 1 CFU) of $10^5$ in the spleen and $10^4$ in the lung. For the TnSeq screen, groups of three to six mice per genotype were infected, including 52 CC strains, 8 parental strains, and single-gene knockout mice (*Nos2$^{-/-}$*, *Cybb$^{-/-}$*, *Ifnγ$^{-/-}$* and *Rag*N12). Mice were infected over three infection batches, as denoted in *Figure 1—source data 1*. Burden and immunological data from all surviving animals are provided in *Figure 1—source data 1*. At indicated time points mice were euthanized, and organs were harvested then homogenized in a FastPrep-24 (MP Biomedicals). CFU was determined by dilution plating on 7H10 agar with 20 µg/mL kanamycin. For library recovery, approximately $1 \times 10^6$ CFU per mouse was plated on 7H10 agar with 20 µg/mL kanamycin. After three weeks of growth, colonies were harvested by scraping and genomic DNA was extracted. The relative abundance of each transposon mutant was estimated as described (*Long et al., 2015*).

Single strain validation aerosol infections were performed in a Glas-Col machine to deliver 50–150 CFU/mouse. At indicated time points, mice were euthanized, and organs were harvested then homogenized in a FastPrep-24 (MP Biomedicals). CFU was determined by dilution plating on 7H10 agar with 20 ug/mL kanamycin or 50 µg/mL hygromycin as required.

Chromosomal equivalent (CEQ) was enumerated according to previously published protocol (*Lin et al., 2014*; *Munoz-Elías et al., 2005*). Cytokines and chemokines were assayed from organ homogenates using the pro-inflammatory focused 32-plex (Eve Technologies, Calgary, CA).

For pooled mutant infections, three to five mice per genotype (B6, CC051, PWK, CC042, CC005, CC018) were infected with a pool of deletion mutants at equal ratios via the intravenous route (1 $\times 10^6$ CFU/mouse resulting in an infectious dose (D1 CFU) of $1 \times 10^5$ in the spleen). At indicated time points, approximately 10,000 CFU from the spleen homogenate of each mouse was plated on 7H10 agar. Genomic DNA was extracted for sequencing as described previously (*Long et al., 2015*). Sequencing libraries spanning the variable region of each q-Tag were generated using PCR primers binding to regions common among all q-Tags, similar to previously described protocols (*Bellerose et al., 2020*; *Blumenthal et al., 2010*; *Martin et al., 2017*). In each PCR, a unique molecular counter was incorporated into the sequence to allow for the accurate counting of input templates and account for PCR jackpotting. The libraries were sequenced to 1000-fold coverage on an Illumina NextSeq platform using a 150-cycle Mid-Output kit with single-end reads. Total abundance of each mutant in the library was determined by counting the number of reads for each q-Tag with a unique molecular counter. Relative abundance of each mutant in the pool was then calculated by dividing the total abundance of a mutant by the total abundance of reads for wild-type H37Rv. The relative abundance was normalized to relative abundance at initial infection (Day 0) and log$_2$ transformed. Fitness was calculated as previously described (*Palace et al., 2014*). Burden and normalized counts from all *Mtb* mutants in each mouse are provided in *Figure 4—source data 2*.

For CC030 x CC029 F$_2$ infections, 251 F$_2$ mice (including equivalent numbers of male and female mice) were infected via IV route with an infectious dose of $10^5$ CFU of TnSeq library (as described above), to replicate the original CC infection experimental conditions. Mice were sacrificed at 1 month post-infection and bacterial burden was quantified by plating for CFU (as described above).

## Quantification and statistical analysis

### TnSeq analysis

TnSeq libraries were prepared and counts of each transposon mutant were estimated as described (*Long et al., 2015*). NCBI Reference Sequence NC_018143.1 was used for H37Rv genome and annotations. A total of 123 libraries were sequenced, capturing 60 distinct mouse genotypes. In the majority of cases, two replicate mouse libraries were used per genotype. Only a single TnSeq library was obtained for CC010, CC031, CC037, CC059, CC016, and PWK/PhJ. Insertion mutant counts across all libraries were normalized by beta-geometric correction (*DeJesus et al., 2015*), binned by gene, and replicate values for each mouse genotype averaged. Mean values for each gene were

divided by the grand mean then $\log_2$ transformed and quantile normalized. The resulting phenotype values were used for both WGCNA and QTL mapping.

To eliminate genes having no meaningful variation across the mouse panel, statistical tests of $\log_2$ fold change (LFC) in counts between all possible pairs of mouse genotypes were performed by resampling (*DeJesus et al., 2015*). 679 'significantly varying genes' were identified whose representation varied significantly (FDR < 5%) in at least two independent comparisons. For relative mutant abundance estimates, LF C in counts between in vitro-grown H37Rv (six replicate libraries) vs libraries from each mouse genotype were determined by resampling as above. LFC, Q-values and modules for TnSeq data across the mouse strains is available in *Figure 4—source data 1*.

## WGCNA analysis

Weighted gene correlation network analysis (WGCNA) was applied to categorize the 679 significantly varying genes into 20 internally-correlated modules (*Langfelder and Horvath, 2008*). Modules were filtered (intramodular connectivity >0.6) to obtain the most representative genes. First principal component scores of module eigengenes were used as phenotype values for QTL mapping after first winsorizing (q = 0.05) using the R package *broman* (https://cran.r-project.org/web/packages/broman/index.html).

In order to perform association analysis between modules of genes and clusters of mice (*Figure 5B*), the mice were clustered based on the matrix of TnSeq LFCs for significantly varying genes using *hclust* in R (with the 'Ward.D2' distance metric). Then, for each module of genes, the LFCs in each cluster of mice were pooled and compared to all the other mice using a t-test, identifying modules with a mean LFC in a specific mouse cluster that is significantly higher or lower than the average across all the other mice. The resulting p-values over all combinations of gene modules and mouse clusters were adjusted using Benjamini-Hochberg for an overall FDR < 0.05.

## Disease-related trait analysis and heritability estimation

For the trait heatmap, trait values were clustered (*hclust* in R package *heatmaply*; traits scaled as per default function) and dendrogram nodes colored by 3 k-means. Correlation between disease-related TB traits for both IV and aerosol validation experiments was determined by Pearson's correlation and visualized using *corrplot* (ordered by *hclust*) (*Figure 1—figure supplements 1 and 2*). Heritability ($h^2$) of the immunological and TB disease-related traits was calculated by estimating the percent of variation attributed to between strain differences relative to within strain noise as previously described (Appendix1) (*Noll et al., 2020*). This is explicitly: SS(strain)/SS(total) in an ANOVA table (where SS(total) is SS(strain)+ SS(error)) (SS; sum of squares). p-Values were calculated by ANOVA and multiple-test corrected using the Benjamini-Hochberg method. Throughout the text, correlations are cited using the following standardized nomenclature: 0–0.19 = very weak, 0.2–0.39 = weak, 0.40–0.59 = moderate, 0.6–0.79 = strong, 0.8–1.0 = very strong correlation.

## Genotyping and QTL mapping

A subset of the inbred CC mice used in the analysis were genotyped on the GigaMUGA array (*Morgan et al., 2015*) available from Neogen Inc The inbred parents, $F_1$s and $F_2$ mice from the CC030xCC029 cross were genotyped on the MiniMUGA array (*Sigmon et al., 2020*) at Neogen Inc, For CC030 x CC029 $F_2$ QTL analysis, markers were filtered to 2499 markers that differentiated between CC029 and CC030 haplotypes (*Figure 3—source data 1*). For QTL mapping in the $F_2$ panel, genotype (*Figure 3—source data 1*) and lung burden data (*Figure 3—source data 2*) from 251 *Mtb*-infected $F_2$ individuals was imported into R (version 3.6.1) and formatted for R/qtl2 (version 0.20) (*Broman et al., 2019*). QTL mapping incorporated kinship as a covariate using the LOCO (Leave One Chromosome Out) method. Further, sex and infection batch were also considered as covariates for mapping. LOD scores were calculated within R/qtl2 to assess genotypic associations with lung burden at each marker. QTL significance thresholds were established by 10,000 permutations.

For QTL mapping in the CC panel, the Most Recent Common Ancestor (*Srivastava et al., 2017*) 36-state haplotypes were downloaded from the UNC Systems Genetics Core Facility and simplified to 8-state haplotype probabilities (for the 8 CC founder strains), which is appropriate for additive genetic mapping. We generated 36-state haplotype probabilities from the individual CC mice genotyped on GigaMUGA and combined these data with the MRCA data to obtain a common genome cache.

For CC QTL analysis, genotype and phenotype data were imported into R (version 3.6.1) and reformatted for R/qtl2 (version 0.20) (*Broman et al., 2019*). Individual TnSeq and clinical trait phenotype values were winsorized (q = 0.006) as above. GigaMUGA annotations were downloaded from the Jackson Laboratory, and markers were thinned to a spacing of 0.1 cM using the reduce_markers function of R/qtl2. The final genetic map contained 10,067 markers. QTL mapping was carried out using a linear mixed model with LOCO (leave one chromosome out) kinship. For clinical trait scans, batch (denoted by 'block' in *Figure 1—source data 1*) was included as an additive covariate. Significance thresholds for QTL were estimated using 10,000 permutations (scan1_perm function). For each trait, the maximum LOD scores from the permutation scans were used to fit generalized extreme value distributions, from which genome-wide permutation p-values were calculated. LOD profiles and effect plots were generated using the plotting functions of the R/qtl2 package. Multiple QTL at similar genetic locations were assessed for independence using qtl2pleio with 400 bootstrap samples (*Boehm et al., 2019*). The quantile-based permutation thresholding method of *Neto et al., 2012* was used to assess the statistical significance of co-mapping traits. The NL-method, which determines the LOD thresholds controlling genome-wide error rate for a given p-value and 'hotspot' size, was employed.

## Candidate gene prioritization

To identify potential candidate genes, we focused on three QTL that were either statistically significant (*Tip5*, *Hip42*) or were validated by intercross (*Tip8*). For each QTL interval (determined by Bayesian interval in qtl2), we identified mouse genes that were in differentially expressed modules between infected lungs of resistant and susceptible mouse strains (*Moreira-Teixeira et al., 2020*). Of these genes, we next used the Sanger sequence data (*Keane et al., 2011*) to filter on genetic variants segregating between CC founder haplotypes. Where there were clear causal haplotypes, we further filtered to genes with missense or nonsense variants.

## Acknowledgements

We thank all members of Sassetti Lab, past and present for technical help and discussions; Dr. Nathan Hicks and Dr. Sarah Fortune for kindly providing the *rnaseJ* mutant; Dr. David Tobin for insightful manuscript comments; Dr. Dennis Ko for QTL acronym creativity; and the Systems Genetic Core at UNC for their help in procuring CC mice in timely fashion. This work was supported by NIH grants AI132130 to C M Sassetti and FPMV; U19AI100625 to FPMV and MF; a fellowship from the Charles H King Foundation to C M Smith; and a HHMI Gilliam Fellowship A20-0146 to BKH. The genetic characterization of the CC strains was supported in part by NIH grant U24HG010100 to FPMV.

## Additional information

### Funding

| Funder | Grant reference number | Author |
| --- | --- | --- |
| National Institute of Allergy and Infectious Diseases | AI132130 | Fernando Pardo-Manuel de Villena Christopher M Sassetti |
| National Institute of Allergy and Infectious Diseases | U19AI100625 | Fernando Pardo-Manuel de Villena Martin T Ferris |
| Howard Hughes Medical Institute | A20-0146 | Brea K Hampton |
| National Human Genome Research Institute | U24HG010100 | Fernando Pardo-Manuel de Villena |
| Bank of America | Charles H King Postdoctoral Fellowship | Clare M Smith |

| Funder | Grant reference number | Author |
| --- | --- | --- |

The funders had no role in study design, data collection and interpretation, or the decision to submit the work for publication.

## Author contributions

Clare M Smith, Conceptualization, Formal analysis, Investigation, Methodology, Supervision, Validation, Visualization, Writing – original draft; Richard E Baker, Data curation, Formal analysis, Methodology, Visualization, Writing – original draft; Megan K Proulx, Formal analysis, Investigation, Validation, Writing – review and editing; Bibhuti B Mishra, Jarukit E Long, Michelle M Bellerose, Andrew J Olive, Charlotte J Reames, Brea K Hampton, Colton L Linnertz, Ginger D Shaw, Pablo Hock, Timothy A Bell, Investigation, Writing – review and editing; Sae Woong Park, Ha-Na Lee, Michael C Kiritsy, Investigation, Validation, Writing – review and editing; Kenan C Murphy, Methodology, Validation, Writing – review and editing; Kadamba Papavinasasundaram, Sabine Ehrt, Dirk Schnappinger, Supervision, Validation, Writing – review and editing; Frederick J Boehm, Investigation, Methodology, Writing – review and editing; Rachel K Meade, Methodology, Writing – review and editing; Fernando Pardo-Manuel de Villena, Funding acquisition, Resources, Supervision, Writing – review and editing; Martin T Ferris, Funding acquisition, Investigation, Methodology, Supervision, Writing – review and editing; Thomas R Ioerger, Investigation, Methodology, Resources, Supervision, Writing – original draft; Christopher M Sassetti, Conceptualization, Formal analysis, Funding acquisition, Project administration, Supervision, Writing – original draft

## Author ORCIDs

Clare M Smith http://orcid.org/0000-0003-2601-0955
Megan K Proulx http://orcid.org/0000-0002-9524-8302
Bibhuti B Mishra http://orcid.org/0000-0002-7203-1653
Sae Woong Park http://orcid.org/0000-0002-4649-4566
Ha-Na Lee http://orcid.org/0000-0003-4136-0128
Michael C Kiritsy http://orcid.org/0000-0001-8364-8088
Michelle M Bellerose http://orcid.org/0000-0003-0232-9953
Andrew J Olive http://orcid.org/0000-0003-3441-3113
Kenan C Murphy http://orcid.org/0000-0002-3677-2876
Kadamba Papavinasasundaram http://orcid.org/0000-0001-7837-1344
Frederick J Boehm http://orcid.org/0000-0002-1644-5931
Charlotte J Reames http://orcid.org/0000-0001-5579-5881
Rachel K Meade http://orcid.org/0000-0002-8322-0257
Brea K Hampton http://orcid.org/0000-0001-7167-5652
Colton L Linnertz http://orcid.org/0000-0003-2969-8193
Ginger D Shaw http://orcid.org/0000-0003-2590-4973
Timothy A Bell http://orcid.org/0000-0002-9546-6334
Sabine Ehrt http://orcid.org/0000-0002-7951-2310
Fernando Pardo-Manuel de Villena http://orcid.org/0000-0002-5738-5795
Martin T Ferris http://orcid.org/0000-0003-1241-6268
Christopher M Sassetti http://orcid.org/0000-0001-6178-4329

## Ethics

Mouse studies were performed in strict accordance using the recommendations from the Guide for the Care and Use of Laboratory Animals of the National Institute of Health and the Office of Laboratory Animal Welfare. Mouse studies at the University of Massachusetts Medical School (UMASS) were performed using protocols approved by the UMASS Institutional Animal Care and Use Committee (IACUC) (Animal Welfare Assurance Number A3306-01) in a manner designed to minimize pain and suffering in Mtb-infected animals. Any animal that exhibited severe disease signs was immediately euthanized in accordance with IACUC approved endpoints. All mouse studies at UNC (Animal Welfare Assurance #A3410-01) were performed using protocols approved by the UNC Institutional Animal Care and Use Committee (IACUC).

## Decision letter and Author response

Decision letter https://doi.org/10.7554/eLife.74419.sa1
Author response https://doi.org/10.7554/eLife.74419.sa2

## Additional files

### Supplementary files
• Transparent reporting form

### Data availability
All relevant data to support the findings of this study are located within the paper and supplementary files. Genome sequence data is deposited in the NCBI Gene Expression Omnibus (GEO), accession number GSE164156. All raw phenotype values and QTL mapping objects are located on GitHub @sassettilab in the https://github.com/sassettilab/Smith_et_al_CC_TnSeq, (copy archived at swh:1:rev:2ded9735b23d9780eb7872eb55625cff35090430) repository.

The following dataset was generated:

| Author(s) | Year | Dataset title | Dataset URL | Database and Identifier |
|-----------|------|---------------|-------------|-------------------------|
| Smith CM | 2021 | Host-pathogen genetic interactions underlie tuberculosis susceptibility in genetically diverse mice | http://www.ncbi.nlm.nih.gov/geo/query/acc.cgi?acc=GSE164156 | NCBI Gene Expression Omnibus, GSE164156 |

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

## Appendix 1

**Appendix 1—table 1.** Heritability ($h^2$) estimates for each measured TB-disease associated phenotype (Tuberculosis ImmunoPhenotypes; *Tip*).

$h^2$ was calculated from the percentage of variation attributed to strain differences in each trait across the CC strains, as previously described (***Noll et al., 2020***). P-values were calculated by ANOVA and multiple-test corrected using the Benjamini-Hochberg method. Weight change is the percentage of weight (grams), CFU/CEQ is $log_{10}$ transformed, cytokines are measured in pg/mL lung homogenate and $log_{10}$ transformed.

| Trait | $h^2$ (%) | p-value | Adj. p-value |
|---|---|---|---|
| IFN-γ | 87.70 | 7.90E-20 | 2.21E-18 |
| Lung CFU | 83.30 | 5.83E-15 | 8.16E-14 |
| Lung CEQ | 80.55 | 2.65E-14 | 2.47E-13 |
| CXCL1 | 81.57 | 8.19E-14 | 5.73E-13 |
| MIG | 81.19 | 1.62E-13 | 9.09E-13 |
| MIP-2 | 80.82 | 3.07E-13 | 1.43E-12 |
| IP-10 | 80.13 | 9.73E-13 | 3.89E-12 |
| M-CSF | 79.33 | 3.54E-12 | 1.24E-11 |
| IL-17 | 78.85 | 7.43E-12 | 2.31E-11 |
| MIP-1α | 78.02 | 2.53E-11 | 7.08E-11 |
| G-CSF | 77.71 | 3.98E-11 | 1.01E-10 |
| MCP-1 | 77.05 | 1.00E-10 | 2.34E-10 |
| IL-1α | 75.62 | 6.60E-10 | 1.42E-09 |
| IL-6 | 73.97 | 4.89E-09 | 9.77E-09 |
| RANTES | 73.70 | 6.63E-09 | 1.20E-08 |
| Spleen CFU | 72.94 | 6.83E-09 | 1.20E-08 |
| LIF | 73.48 | 8.50E-09 | 1.40E-08 |
| VEGF | 73.08 | 1.34E-08 | 2.08E-08 |
| IL-1β | 71.66 | 6.11E-08 | 9.00E-08 |
| Weight Change | 67.56 | 8.75E-07 | 1.22E-06 |
| MIP-1β | 68.51 | 1.23E-06 | 1.65E-06 |
| TNF-α | 66.38 | 7.39E-06 | 9.41E-06 |
| LIX | 65.30 | 1.70E-05 | 2.07E-05 |
| Eotaxin | 64.27 | 3.63E-05 | 4.23E-05 |
| IL-10 | 63.97 | 4.48E-05 | 5.02E-05 |
| IL-2 | 62.87 | 9.59E-05 | 1.03E-04 |
| Spleen CEQ | 60.43 | 1.09E-04 | 1.13E-04 |
| IL-9 | 56.79 | 3.19E-03 | 3.19E-03 |

