## [Editor Report]

This work takes advantage of the genetic diversity of a panel of mice, termed the collaborative cross, to identify those host factors that contribute to heterogeneous outcomes after tuberculosis infection. The authors infect this panel of mouse strains with pools of *Mycobacterium tuberculosis* transposon mutants, allowing identification of specific host genotypes that confer fitness effects on certain bacterial mutants. The resulting analyses identify loci that affect quantitative immunological phenotypes or fitness of select bacterial mutants. The study is likely to be an important resource to microbiologists in general and those individuals studying the host immune response to tuberculosis infection

---

## [Decision Letter]

**Decision letter after peer review:**

[Editors’ note: the authors submitted for reconsideration following the decision after peer review. What follows is the decision letter after the first round of review.]

Thank you for submitting your work entitled "Host-pathogen genetic interactions underlie tuberculosis susceptibility" for consideration by *eLife*. Your article has been reviewed by 5 peer reviewers, and the evaluation has been overseen by a Reviewing Editor and a Senior Editor. The reviewers have opted to remain anonymous.

We are sorry to say that, after consultation with the reviewers, we have decided that your work will not be considered further for publication by *eLife*.

Specifically, reviewers concur that your manuscript reports some interesting data. The application of the CC to study the host immune response to TB will yield useful findings. However, reviewers raised concerns regards the statistical approach taken to assess differences in phenotypes/outcomes between CC strains. The primary concern is that the numbers of animals per group sampled is small. When this is combined with the expected heterogeneity in the data, the statistical comparisons as reported are difficult to interpret. After extensive discussions, we were unable to come to a clear sense of whether the observations are robust. Based on this we have opted to reject the manuscript in its current form. That said, the manuscript does report some interesting data that would constitute a useful resource for the field. Hence, we would be willing to consider it again, if you feel you can address the comments. In this context, reviewers have made some extensive recommendations. Please consider these and the suggestions for statistical analyses. After reconsidering, the aerosol infection experiments suggested by reviewer 2 would not be necessary in a revised version, assuming that you can provide reassurance as to the robustness of the findings by clarifying or revising statistical methods. Acceptance of the revised manuscript is not guaranteed

*Reviewer #1:*

In two previous papers (2016 and 2019, mBIO), Smith et al. have used collaborative cross mice to better understand host susceptibility to TB. Those papers used aerosol infection.

In this work, the authors use intravenous infections so as to be able to test a library of bacterial mutants that the lab has previously constructed with the view of identifying the comprehensive set of virulence genes required for infection. the goal of this work would be to understand better the differences in individuals who are more or less resistant to TB and how the pathogen's wider gene set allows it to infect a greater proportion of people successfully.

In two previous papers (2016 and 2019, mBIO), Smith et al. have used collaborative cross mice to better understand host susceptibility to TB. Those papers used aerosol infection.

In this work, the authors use intravenous infections so as to be able to test a library of bacterial mutants that the lab has previously constructed. The ability to use large numbers of bacteria delivered intravenously should theoretically avoid the bottlenecks resulting from aerosol infection. However, possibly because of the small numbers of mice used from each genotype (2-6, average 3), the data are noisy. This is obvious even in Figure 1A where bacterial burdens in only 15/59 groups are statistically different from the B6 mouse used, using relatively nonstringent statistical analyses. This most likely represents experimental variability, which disadvantages the rest of the work and its conclusions.

The problem becomes obvious early in the paper when they look at the correlation between lung and spleen bacterial burdens. An R2 of 0.43 is not a moderate correlation but a poor correlation. Either there is an interesting reason for this or more likely it simply reflects small numbers. The correlations between bacterial burdens and cytokines/chemokines are also week to nonexistent. Here too, the lung – spleen disparity is a problem. For instance, IL-1b and TNFa are weakly correlated for lung burdens and not at all for spleen. For IFNγ, which the authors make a point of detailing, the data do not support their statements. There is no correlation even if the two or three high (#) strains are excluded. This makes the TipQTL analysis of genetic variants that control TB immunophenotypes also suspect and the statistics are weak. The argument that Tip8 is a strong predictor of lower bacterial burden is also not convincing based on Figure 3B. First, there is strong overlap in the bacterial CFU between samples 2 and 3, one of which has the Tip8 CAST and the other which does not. Second, the last group which also contains the CAST at Tip8 looks not different from the third one. What would be the reason for that? Overall, these data are not statistically robust. This looks like noise.

It is also not clear these new findings relate to their previous ones in Smith et al. 2019 mBio. In the current paper, the authors follow up by infecting a B6/CAST cross by the aerosol route to conclude that the bacterial control of CAST mice is independent of IFNγ-secreting cells. In the previous paper, they perform a different cross, CC001/CC042 to also dissociate bacterial control from IFNγ production. How do the current findings relate to the previous ones? The previous ones are far from understood and it feels as if the additional data are more confusing than clarifying.

On this weak framework, the authors try to determine if the CC mice can reveal the extended requirement for pathogen genes that are not reflected in single mutant strains. This work begins in Figure 4. (First, a minor point, I compute the genes in B6 to be 159+53=212 whereas on page 15, line 275 states that they identified 234 genes.) Beyond this, I have several questions:

1) The authors state that while the pathogen gene sets were different from mouse strain to mouse strain, the number of genes required in each strain was roughly the same. What is their explanation/model for this constant number, given a 3-log difference in bacterial burdens between the strains reflecting enormous differences in susceptibility? How could this model be tested? This is an important point.

The authors report a common set of ~ 130 genes. Are there other common subgroups defined by common pathogen gene sets? These types of analyses are essential to both validating the system and in so doing may give biological information. What about the additional genes in the CC subset over the immunodeficient KO set? What can be learned from them?

And beyond this, I do not follow their further analyses and am just not convinced that they are meaningful. There seems to be a lot of statistical massaging with seemingly erudite analyses but with the small number of mice used per genotype, the large error and the glossing over of problems, it is hard to see how this paper represents a significant advance.

I hope that the paper is being co-reviewed by individuals expert in mouse genetics and in statistics or in both. If not, this will be essential.

*Reviewer #2:*

This work takes advantage of the genetic diversity of a panel of mice (called the collaborative cross (CC) mice) to identify mice that produce heterogeneous outcomes after *M. tuberculosis* (Mtb) infection. The hope is to better recapitulate the diversity of outcomes of Mtb infection seen in humans, a current major limitation of the dominant B6 mouse model. While some important human Mtb infection outcomes (most notably, latency) are apparently not captured by the CC mice, there is nevertheless an impressive variation in infection outcomes in the CC panel (>100-fold range in lung CFU burdens, for example). There is also large variation in host responses. For example, CAST mice control infection but are found to lack an IFNγ response thought previously to be critical for Mtb control based on studies in B6 (but known to be similarly absent from some human controllers). The authors also infect the CC mice with pools of transposon mutants, allowing identification of specific host genotypes that confer fitness effects on specific bacterial mutants. The resulting analyses identify loci that affect quantitative immunological phenotypes (TipQTL) or fitness of bacterial mutants (HipQTL). No causative genetic variants underlying these QTL are validated by independently generated mutation, so in some sense, the "discoveries" here are limited. However, the impact of this work is likely to be great as a resource that can be mined by investigators in the field to identify specific mouse strains in which, for example, a specific bacterial mutant has a phenotype, or a particular host response is observed, greatly facilitating followup work.

Given the lack of a major new biological insight or mechanism, perhaps this should be considered as a "Tools and Resources" paper (https://reviewer.elifesciences.org/author-guide/types) rather than a regular research paper.

The use of a high-dose IV model of infection, while necessary for TnSeq experiments, may limit the relevance of the observations to pulmonary TB. This concern is somewhat ameliorated by the fact that the susceptibility of the parent strains largely recapitulates the expected phenotype based on previous aerosol infections. However, for a few key CC mice with interesting phenotypes (e.g., the exacerbated weight loss in CC030 or CC040) it would be nice to validate in a replication experiment and that the phenotype is consistently seen even with aerosol infections. The experiment in Figure 3 could potentially serve as such validation, but it is not clear whether this experiment was using IV or aerosol infection (please specify this clearly in the legend and text regardless).

Line 143: please specify in the text the dose (CFU) of bacteria that were delivered IV. This is important information and should be in the main text.

Line 159: "Thus, the CC panel encompasses a much greater quantitative range of susceptibility than standard inbred lines." – this is not obviously true. The range of CFU budens seen from NOD to WSB appears to capture most of the range of the CC panel. Perhaps change the wording here?

Line 208: is it really "none", as in not a single IFNγ+ T cell? The bar graph suggests that there is some, so perhaps revise this sentence.

Line 251: "mice that contained CAST at Tip8, Tip10 or both loci had reduced CFU burden". It appears from the data in Figure 3 that in fact the only statistically significant difference is for the Tip8 locus, and that the presence of the CAST Tip10 allele abolishes this difference. If that is correct, the manuscript needs to be clear that in fact the Tip10 locus did not validate (statistically).

Line 274: "Consistent with our previous work, we identified 234 Mtb genes that are required for growth or survival in Mtb in B6 mice". In fact, how consistent are the data? Of the 234 genes identified, how many were previously identified? How many of the previously identified genes were in the set of 234? Please specify so that the readers can get a sense of the experiment-to-experiment variability of the approach.

Why would more Mtb genes be essential in Rag1-/- mice as compared to WT mice? This seems unexpected to me and perhaps deserves some comment? What are these genes – could this be discussed at least a little? Line 340 mentions that ESX genes are among those preferentially required in Rag1-/- (though obviously they are also required in B6, so presumably this does not account for the 'additional' genes required in Rag1-/-). Could the authors elaborate a bit more on what they think is going on here?

Line 281: "As more CC strains, and presumably more distinct immune states, were included in the analysis, the cumulative number of genes necessary for growth in these animals also increased". It is possible that repeating the experiment with more and more B6 animals would also increase the cumulative number of genes necessary? In other words, might the sensitivity of the TnSeq method be what is limiting here as opposed to host genetic diversity? I presume the authors have a good argument against this concern; it would be helpful for readers to hear it.

Line 295: "the resulting mini-pool was subjected to in vivo selection in the same manner as the TnSeq study" – does this mean that the minipool was used to infect ALL the same mouse strains (all the CC mice) as in Figure 4A? This isn't clear. A related concern regarding Figure 4C: why are selected mouse strains shown in this figure? Are they the strains that showed the biggest signal in the TnSeq experiment? The worry is they were cherry picked. This could be explained better.

Line 298: "relative abundance predicted by TnSeq was reproduced with deletion mutants" – this is difficult to evaluate because the fitness of each of the mutants in each of the strains in Figure 4A is not stated. I realize the full dataset of all the TnSeq data is available as a spreadsheet but it would be nice if the reader didn't have to dig through that. Ideally it would be great to create a searchable website in which investigators could enter their gene of interest and the strains in which the mutant shows a phenotype (and the magnitude of the phenotype) could be displayed.

Line 312: again, it is not clear what the TnSeq predicted. Please state that result here too.

Line 435: which HipQTL are associated with both ahpC and eccD1? Please specify. I don't see this information in the cited figure either (Figure 4E). Please make sure this is correct.

*Reviewer #3:*

In this elegant and expansive study, the authors have studied bacterial adaptive response in the context of variable immune pressure exerted by diverse hosts. They used the Collaborative Cross (CC) mouse panel in conjunction with a library of Mtb transposon mutants to uncover *M. tuberculosis* genetic requirements with host genetics and immunity. The authors report several new findings. For example, they report that several Mtb virulence pathways become unmasked only under specific host microenvironments. They also found that a number of genotypes were able to control bacterial replication but had very low levels of IFNγ. Also, dependent on the genetic composition of the host, related disease traits such as lung bacterial burden and weight loss could be dissociated. Another significant outcome was identifying candidate genes underlying the QTLs and showing that the QTLs were generally distinct from CC loci that control immunity to other pathogens.

Overall, I find this to be a complete and thorough study.

*Reviewer #4:*

I reviewed this study only to evaluate the robustness of the results and the statistical evaluations used in the study. No attention was paid to novelty or scientific interest. Some of the statistical approaches used by the authors were beyond my expertise and require further analysis by a professional statistician to which I lay no claim to being. This study used multiple simultaneous comparisons of both animal strains and bacterial strains, with what appears to be generally quite small sample size, non-replicated experiments. Data files that are presented either omit entirely some experiments or provide only summary data of mean values without indicating numbers of subjects, measures of variability such as standard deviation; no information on the results of replicate studies, if any were performed, are in these data files. Therefore I find it impossible to determine if many of the "significant" associations are robust. I give some examples below:

1) Data for Figure 1. The data set for figure 1 only summary data. There is no indication of the number of animals studied in each group and no indication of standard deviation. Also, there is no indication in the data set or in the text if any of the experimental results for 1A to 1D were based on replicate studies. The underlying data for figures E to H are not available in the figure 1 data file; the text gives the range of subject sizes and states that two independent experiments were performed, but it is unclear if the data presented in the figures represents both experiments, just one experiment, or if the data presented includes both experiments whether the results were simply combined or if there was some normalization performed. The statistical analysis method does not say whether the data were log-transformed before analysis. All of the author's conclusions may be absolutely correct but it is difficult to know this without more detailed information.

2) Data for Figure 4. The data set for this figure only includes the underlying data for 4A. No underlying data are given for 4C, D and E. No indication is given of the number of repeated experiments, and if these were performed what the results were. No method is given for the statistical analysis used in 4D. The authors conclude that there are significant differences between mouse strains for several Mtb strains, but give no information on the numbers of mice studied, variation within groups, repeat experiments, etc; also no statistical analysis is performed. Again, the author's conclusions may be absolutely correct but it is difficult to know this without seeing the underlying data.

While the authors may believe that their answers to the transparent reporting form were complete and accurate, based on what I have found I'm not certain that this is correct.

*Reviewer #5:*

Sample size and reproducibility. The groups were small (average of 3 mice, sometimes only 2), which could result in a high variance per group. It would be important to evaluate how reproducible are the phenotypic measurements per strain. The authors mention reproducibility per genotype (Page 8 line 177), but they do not clearly show it. Perhaps a dimension reduction approach (PCA or UMAP) could be used to evaluate the distance between replicates. As specific examples: How many animals were used per group in outlier strains (Figure 1 C and D)? What was the variance for the y-axis measurements? Could variance explain the outlier nature of the strain?

QTL analysis. The CC QTL analysis faces similar challenges to genetic studies in humans. Overlooking complex population structure can lead to inflated false positives but can also prevent harvesting the full potential of the samples. Limitations of the QTL analysis are briefly introduced (Page 11 line 241) but the explanation in methods is convoluted and difficult to follow. The study could benefit from using their SNPs data for genome-wide association testing using all strains correcting by population structure. This strategy was shown to be robust when performed to a study of similar framework using different inbred strains and their crosses (PMID 32365090). This method could address sample size and QTL as the animals would be treated as unique samples (not per group) correcting by population structure (similarities between samples in the dataset). If correct, this would be a more straightforward genome-wide analysis.

Detailed comments:

In methods, it is stated that TipQTL significance was assessed using permutations. However, in the Results section and in table 1 it is not clearly stated if the p values are raw or corrected via the permutation.

Line 63: Co-existence of Mtb and *Homo sapiens* is now widely accepted, perhaps 2000-6000 years. Has co-evolution been demonstrated or is this hypothesized? Same point with the role of genetic variation presented as established.

When describing correlations, suggest a standard terminology: Strong, Moderate, Weak. Highly correlated is of uncertain meaning. On line 182 the R2 was 0.57 but the text said correlated. Some use the following: 0-0.19 = very weak, 0.2-0.39 = weak, 0.40-0.59 = moderate, 0.6-0.79 = strong, 0.8-1 = very strong correlation

Line 151. What does it mean when the susceptibility is largely consistent with other studies. Can the authors spell out the inconsistencies somehow?

Line 220. What is reasonable statistical confidence?

Line 272. At some point, it would be useful to read in the text that the combined studies of WT Bl6, RAG^-/-^, IFN-/-, Nos2-/- and CYbb-/- mice produced x essential genes, and that the CC raised it from x to y. The paragraph tells us the 234 genes were identified in Bl6 and that the plateau was ~750. How far from 234 to 750 was reached with the 4 specified knockout mice?

Line 321. I did not understand what was complex about the kinetics. Perhaps this can be clarified?

[Editors’ note: further revisions were suggested prior to acceptance, as described below.]

Thank you for submitting your article "Host-pathogen genetic interactions underlie tuberculosis susceptibility" for consideration by *eLife*. Your article has been reviewed by 5 peer reviewers, and the evaluation has been overseen by a Reviewing Editor and Bavesh Kana as the Senior Editor. The reviewers have opted to remain anonymous.

Reviewers and I concur that your manuscript provides value to the field as a "Tools and Resource" paper for generating new hypotheses. The genotypic diversity explored in the study, together with an assessment of disease pathology/outcomes, provides a useful framework to pose important questions that are currently topical for tuberculosis. That said, it would be important to frame the study as a resource rather than trying to probe specific questions with the current dataset, which can be limited by the depth of data available currently in the work. I encourage you to consider carefully how best to present this to the field in a manner that acknowledges the limitations and but also creates the space for innovative and explorative future work.

To facilitate this, please address the following concerns:

1. For a paper reporting a potentially useful resource, it is imperative that all data be publicly available for further interrogation and exploration. Reviewers expressed concern that this is not the case with your submission. For the GEO link, data appear available at https://www.ncbi.nlm.nih.gov/geo/query/acc.cgi?acc=GSE164156. However, for the GitHub link, the Smith repository was not located. There was only one repository (Bellerose): https://github.com/sassettilab?tab=repositories. Please address this.

2. In lines 145-147, it is stated: Groups of 3-6 male mice per genotype were infected and TB disease-related traits were quantified at one-month post-infection. Data from all surviving animals are provided in Figure 1-sourcedata1. However, the source data file indicates that the N is 1-3 for most of the data, except for two strains, Bl6 and CC010, which are reported to have 6 mice. There are asterisks for significance in only in a minority of the bars (~15) in Figure 1A. What about the remainder? What n's did they have? Too few to determine significance? This should be clarified.

3. Explain why an R value of 0.43 would be considered a moderate correlation, when elsewhere you suggest that there could be biological differences that may make this correlation weak. Similarly, it is often the case that in inflammatory processes, TNF and IFNγ track together, even if made by different cells, which is not even strictly the case. These are two examples of continued disagreement. Please reconsider carefully how you present these data.

4. Line 184-188. The "significant" associations described in this section are for non-replicate studies with N=2 (CC30, 40 and A/J). Perhaps reconsider the use of "significant" in this context. In Line 253, this experiment used a large N and hence likely produced robust results.

5. Figure 1 A. Giving results of an ANOVA with post-hoc analysis comparison of means to a single mouse strain (C57/BL6) is difficult to interpret because of the potential bias from post-hoc decisions about which strain serves as a comparator. For example, using AJ as a comparator shows no significant differences with other strains. For this reason, excluding this analysis is warranted.

6. Line 199: "A number of genotypes were able to control bacterial replication yet had very low levels of IFNΓ (Figure 1D)" -- for the benefit of the reader, please specify which genotypes fall into this category. Ideally this would be specifically mentioned in the text and in addition the genotypes of interest would be labelled in the figure. Similarly for Figure 1C, it would be helpful if the significant "outlier" genotypes were labelled in the figure so the reader can know which these are.

7. Line 487: "Two TipQTL overlapped with HipQTL (Figure 7)…". Please specify in the text which overlapping QTL are being referred to here. It seems evident from the figure but specifying would assist the reader here.

8. Some of the type on the figures is really tiny, e.g., 1A, 1B, most of figure 4, parts of figure 6, etc. Enlarge the panels with the tiniest type so that they are more readable.

*Reviewer #1:*

The main criticism of the initial submission was the small sample size per CC genotype. Admittedly, and as expected, a number of animals within the susceptible CC genotype died during the study. For this reason, 23 out of the 60 CC genotypes had less than three mice per group, including 3 out of the 4 top CC strains (CC032, CC037, and CC027). In this revised version of the manuscript, the authors addressed the criticism by replicating the difference in lung CFU, weight, and selected cytokines between B6 and the top four CC strains. Moreover, based on the response to reviewer 1 they have indicated robust literature supporting that sample size per group did not impact the statistical power of their analyses. As far as I am concerned, they have addressed most of the reviewer's comments properly. Some of the analyses suggested in the first round of review were not formally evaluated. However, given that the paper was now submitted as a "Tools and Resources" it should be possible for independent groups to explore the data produced by Smith, C. et al. in details or using different analytical approaches. I have not further comments

*Reviewer #2:*

Overall I remain enthusiastic about this manuscript. It has been a longstanding question whether the diversity of immune responses seen in humans could be modeled in mice and this manuscript represents the most substantial (by far) attempt to address this question. Although some key human phenotypes (e.g., "latency") do not appear to be recapitulated in the CC mouse panel, there are definitely several interesting phenotypes reported that are not seen in the typically used B6 strain. For example, the identification of strains of mice that control TB despite low levels of IFNγ production is an interesting phenotype that may phenocopy the "resister" phenotype observed in some humans.

I think the quality of the work is overall high and that, given experimental constraints, sufficient effort was expended to assure the rigor and reproducibility of the datasets.

Originally my main concern was that the paper reports little new in the way of biological insight -- for example, the host genes underlying the TipQTL or HipQTL were not identified, and the underlying mechanisms responsible for the observed immune phenotypes were not dissected. Despite this, I think the manuscript is an important contribution if published as a "Tools and Resource" paper, which is an idea the authors appear to have embraced in their response to reviews. In particular, identification of CC strains which give phenotypes for hundreds of specific bacterial mutants (that give no phenotype in the standard B6 strain) is likely to be extremely helpful to investigators in the field.

1) Please confirm that this manuscript is now submitted as a "Tools and Resource" paper.

2) Line 199: "A number of genotypes were able to control bacterial replication yet had very low levels of [IFNγ] (Figure 1D)" -- for the benefit of the reader, please specify which genotypes fall into this category. Ideally this would be specifically mentioned in the text and in addition the genotypes of interest would be labelled in the figure. Similarly for Figure 1C, it would be helpful if the significant "outlier" genotypes were labelled in the figure so the reader can know which these are.

3) Line 487: "Two TipQTL overlapped with HipQTL (Figure 7)…". Please specify in the text which overlapping QTL are being referred to here. I think it is evident from the figure but I would be reassured I am interpreting the figure correctly if you could be specific here.

4) Some of the type on the figures is absurdly tiny, e.g., 1A, 1B, most of figure 4, parts of figure 6, etc. I would suggest enlarging the panels with the tiniest type so that they are more readable.

*Reviewer #4:*

The sole purpose of my review was to determine if the described experiments were robust, in terms of sample sizes and replication.

The data provided in the manuscript are not robust, with a few exceptions. The experiment studying 60 different mouse strains analyzed for 21 different outcomes (1260 total possible comparisons) utilized a single experiment with mouse strain numbers ranging from 1 to 6 (not 3 to 6 as specified in the manuscript), where the N was 1 for one mouse strain, 2 for 22 strains, 3 for 35 strains and 6 for 2 strains. No replicate studies were performed for the entire group. The chances of analytic variability and assignment of significance to interactions between mouse strain and measured outcome are far too great to regard these data as anything other than hypothesis generating data requiring confirmation. The authors say that the experiment represented in figures 1E-H validate their global conclusions for this data set, whereas they really only validate results for five different mouse strains studied for four different outcomes, two of which are highly correlated (lung cfu and weight loss). This validation study was not repeated, to demonstrate reproducibility, which would bolster those results.

There continues to be a lack of raw data availability supporting some of the various experiments presented in the manuscript. The Github file specified in the manuscript is either not yet posted or not available publicly (and therefore not to me). The data underlying the experiments presented in figure 1 is available only in summary form (mean, N, SD) but not for each mouse studied. Neither the raw nor summary data for figure 4D are not found in any of the data files, and specifically not in the Figure 4 supplemental data 2 file.

Specific comments

1) 184-188. The "significant" associations described in this section are for non-replicate studies with N=2 (CC30, 40 and A/J).

2) 253. This experiment used a large N and hence likely produced robust results.

3) Figure 1 A. Giving results of ANOVA with post-hoc analysis comparison of means to results to a single mouse strain (C57/BL6) is difficult to interpret because of the potential bias from post-hoc decisions about which strain serves as a comparator. For example, using AJ as a comparator shows no significant differences with other strains. For this reason, leaving out that analysis is warranted.

4) Figure 4 C data. Multiple comparisons are performed without apparent adjustment for such, for small N (3 to 5) non-replicate studies. Were these comparisons preplanned?

5) Figure 4D data. No raw data are available.

*Reviewer #5:*

1. In lines 145-147, the authors state: Groups of 3-6 male mice per genotype were infected and TB disease-related traits were quantified at one-month post-infection. Data from all surviving animals are provided in Figure1-sourcedata1.

I am puzzled by the 3-6 mice reported because my reading of the source data file is that the N is 1-3 except for two strains at most, Bl6 and CC010 which are reported to have 6 mice.

There are asterisks for significance in only in a minority of the bars (~15) in Figure 1A. What about the remainder? What n's did they have? Too few to determine significance?

2. It is puzzling why the authors would continue to defend their characterization of an R value of 0.43 as a moderate correlation, when they themselves point out that there could be biological differences that may make this correlation weak. Similarly, it is often the case that in inflammatory processes, TNF and IFNγ track together, even if made by different cells, which is not even strictly the case. These are two examples of continued disagreement.

3. The authors seem to be interested in converting this paper to a Tool/Resource. However, the narrative remains the same as when they wrote it as a Research Paper. For the paper to be considered as a Tool/Resource, the paper needs to be written as such, backing off from strong claims and rewritten as a hypothesis generating screen.

---

## [Author Response]

[Editors’ note: the authors resubmitted a revised version of the paper for consideration. What follows is the authors’ response to the first round of review.]

Reviewer #1:In two previous papers (2016 and 2019, mBIO), Smith et al. have used collaborative cross mice to better understand host susceptibility to TB. Those papers used aerosol infection.In this work, the authors use intravenous infections so as to be able to test a library of bacterial mutants that the lab has previously constructed with the view of identifying the comprehensive set of virulence genes required for infection. The goal of this work would be to understand better the differences in individuals who are more or less resistant to TB and how the pathogen's wider gene set allows it to infect a greater proportion of people successfully.In this work, the authors use intravenous infections so as to be able to test a library of bacterial mutants that the lab has previously constructed. The ability to use large numbers of bacteria delivered intravenously should theoretically avoid the bottlenecks resulting from aerosol infection. However, possibly because of the small numbers of mice used from each genotype (2-6, average 3), the data are noisy.

While this characterization of our group size is correct, we edited the text to make it clear that groups of 3-5 mice per genotype were infected (Line 145). Of the 60 strains, a small number of animals within susceptible CC genotype groups died during the study and were not phenotyped. Data from all surviving animals is provided in Figure1-SourceData1.

This is obvious even in Figure 1A where bacterial burdens in only 15/59 groups are statistically different from the B6 mouse used, using relatively nonstringent statistical analyses. This most likely represents experimental variability, which disadvantages the rest of the work and its conclusions.

We disagree that our data are “noisy”. In general, our ability to detect statistically significant differences in lung CFU between genotypes that vary by ~1 log is comparable to many other studies. Regardless, this is not a point to belabor, as this critique seems to reflect expectations that our study was never designed to fulfill. A classical “mouse study” is powered to detect even small differences in pairwise comparisons between groups. Our study was not designed for the pairwise comparison of genotypes. Instead, this is primarily a genetic mapping study, where the statistical power is derived from the inclusion of multiple genotypes that share a trait-associated haplotype (seven on average in our population). The proper estimate of “noise” in this type of study is based on the heritability of the variation (i.e. the proportion of the trait variation that can be attributed to genetics). Heritability estimates were provided in Figure1-FigureSupp3 of the original submission and show that greater than 80% of the variation in lung CFU and CEQ are due to genetic variation – not “noise”. We also note that the majority of similar genetic mapping studies include only a single animal per genotype. Our strategy to include multiple animals per genotype improves trait value estimation and the overall rigor and reproducibility of our work, relative to almost any other comparable published genetic study. This design is based on systematic studies by members of our group showing that the inclusion of additional CC strains improved mapping power more than additional replicates (Keele, Crouse et al. 2019) and that individual outlier CC strains were identifiable and statistically significant using as few as 2 mice per genotype. Using only 2 mice per strain, CC studies are powered equivalently to a ~400 outbred mouse study per trait (Keele, Zhang et al. 2021). Thus, far from being a limitation of our work, the inclusion of 3-5 animals per genotype makes our study among the most rigorous to date.

The problem becomes obvious early in the paper when they look at the correlation between lung and spleen bacterial burdens. An R2 of 0.43 is not a moderate correlation but a poor correlation. Either there is an interesting reason for this or more likely it simply reflects small numbers. The correlations between bacterial burdens and cytokines/chemokines are also week to nonexistent. Here too, the lung – spleen disparity is a problem. For instance, IL-1b and TNFa are weakly correlated for lung burdens and not at all for spleen.

This comment appears to be based on the reviewer’s own expectations for correlated traits. We both fundamentally disagree with these expectations, and suggest that the reviewer did not consider correlated traits for which there is a much more solid rationale. For example, the reviewer ignores the strong correlation between CFU and CEQ in both lung and spleen (r=0.8 and 0.88, respectively). Based on the criteria suggested by Reviewer 5, these represent “very strong” correlations that support the accuracy of our bacterial burden estimates and our heritability measures. Thus, the more modest correlation between lung and spleen burdens that the reviewer criticizes can be attributed to an important biological difference that is completely consistent with our current understanding of the differential importance of specific immune mechanisms in lung and spleen (e.g.(Sakai, Kauffman et al. 2016)). Similarly, it is unclear why the reviewer expects IL1-b and TNFa to correlate, as they are produced by different cells as a result of different signaling pathways. Much more reasonable expectations would be a correlation between IL-1a and IL-1b (r=0.84), or between the monocyte/granulocyte chemoattractants, MIP-1b, MIP-1a, MCP1, G-CSF, MIP-2, CXCL1 (r=0.89-0.95). These very strong correlations support the accuracy of our cytokine measurements. We ask the reviewer to reconsider these data with an open mind. If they still concerned, we also provide additional aerosol validation data in the new Figure 1 E-H.

In sum, we strongly disagree with this critique of our experimental design, group sizes, and data quality. However, we concede that our study was not designed for robust pairwise comparisons between strains and agree that additional experimental validation/replication strengthens the work. As a result, we have extensively revised Figure 1 to remove conclusions based on pairwise comparisons. These data have been replaced with a simple replication study, which demonstrates the reproducibility of our data in an aerosol infection model. We hope the reviewer agrees that this improves the rigor of the work, and supports subsequent analyses of the dataset.

For IFNγ, which the authors make a point of detailing, the data do not support their statements. There is no correlation even if the two or three high (#) strains are excluded.

This represents a misunderstanding, as the text clearly stated that IFNγ and CFU were not correlated. Regardless, these data have been removed to simplify the conclusions of Figure 1.

This makes the TipQTL analysis of genetic variants that control TB immunophenotypes also suspect and the statistics are weak.

We again note that QTL mapping is not based on simple pairwise comparisons but draws its statistical power from the presence of multiple strains that share a haplotype at a trait associated locus (on average, 7 strains per haplotype at any specified locus). The majority of previous QTL mapping studies use single animals per genotype, and our inclusion of multiple animals per genotype significantly increases the accuracy of our trait value estimations. Finally, this version of the manuscript, like the first, includes a heritability estimate for every trait. The heritability of most traits is >75%, and we are completely transparent in reporting the traits that are more prone to technical variation, which represent cytokines that are produced at relatively low levels. The text referring to this Figure1-FigureSupp3 table has been edited to accentuate its importance.

The argument that Tip8 is a strong predictor of lower bacterial burden is also not convincing based on Figure 3B. First, there is strong overlap in the bacterial CFU between samples 2 and 3, one of which has the Tip8 CAST and the other which does not. Second, the last group which also contains the CAST at Tip8 looks not different from the third one. What would be the reason for that? Overall, these data are not statistically robust. This looks like noise.

The previous Figure 3B compares the F_2_ offspring of genetically diverse CC strains. The observed variation in susceptibility between individuals in this population is not “noise”, but the result of genetic variation. This is both expected and necessary. In no situation would it be reasonable to expect that the distributions of CFU values between these populations would not “overlap”. Instead, we asked if the effect of the Tip8 locus could be discerned, even in the presence of other genetic variants segregating in the CC. The statistically significant difference identified between these populations represents rigorous support for the Tip8 QTL, akin to a replication cohort in a human GWAS study. To take this one step further, we have since genotyped a total of 261 F_2_ intercross mice that we infected with Mtb and quantified lung burden. We conducted QTL mapping and validated the chromosome 7 locus as a genome-wide significant QTL underlying lung CFU. We present this further validation in the new Figure 3C and text. We note that this kind of rigorous independent validation of a QTL is almost never performed before publication.

It is also not clear these new findings relate to their previous ones in Smith et al. 2019 mBio. In the current paper, the authors follow up by infecting a B6/CAST cross by the aerosol route to conclude that the bacterial control of CAST mice is independent of IFNγ-secreting cells. In the previous paper, they perform a different cross, CC001/CC042 to also dissociate bacterial control from IFNγ production. How do the current findings relate to the previous ones? The previous ones are far from understood and it feels as if the additional data are more confusing than clarifying.

The previous Figure 3B compares the F_2_ offspring of genetically diverse CC strains. The observed variation in susceptibility between individuals in this population is not “noise”, but the result of genetic variation. This is both expected and necessary. In no situation would it be reasonable to expect that the distributions of CFU values between these populations would not “overlap”. Instead, we asked if the effect of the Tip8 locus could be discerned, even in the presence of other genetic variants segregating in the CC. The statistically significant difference identified between these populations represents rigorous support for the Tip8 QTL, akin to a replication cohort in a human GWAS study. To take this one step further, we have since genotyped a total of 261 F_2_ intercross mice that we infected with Mtb and quantified lung burden. We conducted QTL mapping and validated the chromosome 7 locus as a genome-wide significant QTL underlying lung CFU. We present this further validation in the new Figure 3C and text. We note that this kind of rigorous independent validation of a QTL is almost never performed before publication.

On this weak framework, the authors try to determine if the CC mice can reveal the extended requirement for pathogen genes that are not reflected in single mutant strains. This work begins in Figure 4. (First, a minor point, I compute the genes in B6 to be 159+53=212 whereas on page 15, line 275 states that they identified 234 genes.)

We thank the reviewer for noting this mistake. It has been fixed (159+53+1+1 = 214 in Figure 4A).

Beyond this, I have several questions:1) The authors state that while the pathogen gene sets were different from mouse strain to mouse strain, the number of genes required in each strain was roughly the same. What is their explanation/model for this constant number, given a 3-log difference in bacterial burdens between the strains reflecting enormous differences in susceptibility? How could this model be tested? This is an important point.

This is a very interesting effect that has been noted in previous publications. While it is not unreasonable to expect more genes to be necessary for fitness in stressful situations, this expectation is not supported by experimental data. Previous TnSeq studies find that while different gene sets are necessary for optimal growth/survival in resistant and susceptible animals, the size of these gene sets are roughly similar (Mishra, Lovewell et al. 2017). This observation is consistent with in vitro studies, where similarly sized gene sets are necessary for growth in different media conditions (Minato, Gohl et al. 2019). One must conclude that while some genes become more important in resistant animals, others become less so. While it is difficult to speculate further, we have added this concept to the discussion of the revised manuscript (lines 592-599).

The authors report a common set of ~ 130 genes. Are there other common subgroups defined by common pathogen gene sets? These types of analyses are essential to both validating the system and in so doing may give biological information. What about the additional genes in the CC subset over the immunodeficient KO set? What can be learned from them?

Identifying “common subgroups defined by pathogen gene sets” is the sole purpose of Figure 5 and the corresponding 600 words of text (lines 422-481). Validation of the system is confirmed by the identification of known functional pathways in TnSeq-derived gene modules (Figure 5A), and the association of these modules with mouse subgroups (Figure 5B). “Additional genes in the CC subset over the immunodeficient KO set” are presented in Figure 5B, along with functional annotation. While we appreciate that this comment was meant as a constructive suggestion, we do not know how to respond, except to note that this analysis was performed and included.

And beyond this, I do not follow their further analyses and am just not convinced that they are meaningful. There seems to be a lot of statistical massaging with seemingly erudite analyses but with the small number of mice used per genotype, the large error and the glossing over of problems, it is hard to see how this paper represents a significant advance.I hope that the paper is being co-reviewed by individuals expert in mouse genetics and in statistics or in both. If not, this will be essential.

We can only hope the reviewer will reconsider their expectations. Our study was never designed to rely on single pairwise comparisons between genotypes, as a typical “mouse study” would. Instead, our study exceeds the rigor of many comparable genetic mapping studies by including multiple animals per genotype and including independent validation of an important QTL (Figure 3). Our “statistical messaging and seemingly erudite analyses” represent rigorous analytical strategies that are appropriate for our study design, and as cited, represent standards in the field (Churchill, Airey et al. 2004, Ferris, Aylor et al. 2013, Gralinski, Ferris et al. 2015, Graham, Swarts et al. 2017, Lorè, Sipione et al. 2020, Noll, Whitmore et al. 2020). Our mouse group size was entirely appropriate for our goals. We demonstrate that our data are not prone to “large error”. The implication that we are “glossing over problems” is unwarranted and vague. We trust that the reviewer will defer to other referees with expertise in mouse genetics and statistics to critique these aspects of the work.

Reviewer #2:This work takes advantage of the genetic diversity of a panel of mice (called the collaborative cross (CC) mice) to identify mice that produce heterogeneous outcomes after M. tuberculosis (Mtb) infection. The hope is to better recapitulate the diversity of outcomes of Mtb infection seen in humans, a current major limitation of the dominant B6 mouse model. While some important human Mtb infection outcomes (most notably, latency) are apparently not captured by the CC mice, there is nevertheless an impressive variation in infection outcomes in the CC panel (>100-fold range in lung CFU burdens, for example). There is also large variation in host responses. For example, CAST mice control infection but are found to lack an IFNγ response thought previously to be critical for Mtb control based on studies in B6 (but known to be similarly absent from some human controllers). The authors also infect the CC mice with pools of transposon mutants, allowing identification of specific host genotypes that confer fitness effects on specific bacterial mutants. The resulting analyses identify loci that affect quantitative immunological phenotypes (TipQTL) or fitness of bacterial mutants (HipQTL). No causative genetic variants underlying these QTL are validated by independently generated mutation, so in some sense, the "discoveries" here are limited. However, the impact of this work is likely to be great as a resource that can be mined by investigators in the field to identify specific mouse strains in which, for example, a specific bacterial mutant has a phenotype, or a particular host response is observed, greatly facilitating followup work.Given the lack of a major new biological insight or mechanism, perhaps this should be considered as a "Tools and Resources" paper (https://reviewer.elifesciences.org/author-guide/types) rather than a regular research paper.

This is a great idea. We agree completely and have submitted the revised version as a “Tools and Resources” paper.

The use of a high-dose IV model of infection, while necessary for TnSeq experiments, may limit the relevance of the observations to pulmonary TB. This concern is somewhat ameliorated by the fact that the susceptibility of the parent strains largely recapitulates the expected phenotype based on previous aerosol infections. However, for a few key CC mice with interesting phenotypes (e.g., the exacerbated weight loss in CC030 or CC040) it would be nice to validate in a replication experiment and that the phenotype is consistently seen even with aerosol infections. The experiment in Figure 3 could potentially serve as such validation, but it is not clear whether this experiment was using IV or aerosol infection (please specify this clearly in the legend and text regardless).

In response to this comment, as well as those of Reviewer 1, we have extensively revised Figure 1 to focus on experimental replication in an aerosol model and increase the value of the larger dataset. The new validation data using an aerosol infection model are presented in Figure 1 E-H. The legend to Figure 3 was edited as requested.

Line 143: please specify in the text the dose (CFU) of bacteria that were delivered IV. This is important information and should be in the main text.

This has been added (lines 143 in main text; also expanded in methods sections, lines 701-708).

Line 159: "Thus, the CC panel encompasses a much greater quantitative range of susceptibility than standard inbred lines." – this is not obviously true. The range of CFU budens seen from NOD to WSB appears to capture most of the range of the CC panel. Perhaps change the wording here?

Absolutely correct. The text has been edited (lines 151-155).

Line 208: is it really "none", as in not a single IFNγ+ T cell? The bar graph suggests that there is some, so perhaps revise this sentence.

These data were removed in response to Reviewer 1.

Line 251: "mice that contained CAST at Tip8, Tip10 or both loci had reduced CFU burden". It appears from the data in Figure 3 that in fact the only statistically significant difference is for the Tip8 locus, and that the presence of the CAST Tip10 allele abolishes this difference. If that is correct, the manuscript needs to be clear that in fact the Tip10 locus did not validate (statistically).

This was a very helpful comment. To build on this validation study of lung CFU, we have now generated a total of 251 F2 intercross mice between informative CC strains, genotyped, infected with Mtb, quantified lung burden at one-month post infection, and performed genome-wide QTL mapping. In this rigorous validation study, identified Tip8 as significantly associated with lung CFU, thus validating the Chr7 locus as the main QTL. We expand discussion of Tip10 as suggested (lines 255-271).

Line 274: "Consistent with our previous work, we identified 234 Mtb genes that are required for growth or survival in Mtb in B6 mice". In fact, how consistent are the data? Of the 234 genes identified, how many were previously identified? How many of the previously identified genes were in the set of 234? Please specify so that the readers can get a sense of the experiment-to-experiment variability of the approach.

The most comparable previous dataset was generated in BALB/c mice. The text now reports that 87% of the genes we report in B6 were also found in BALB/c, and nearly 100% of these genes were consistently identified in the larger CC panel (lines 290-296). These observations support both the sensitivity and specificity of our TnSeq analysis.

Why would more Mtb genes be essential in Rag1-/- mice as compared to WT mice? This seems unexpected to me and perhaps deserves some comment? What are these genes – could this be discussed at least a little? Line 340 mentions that ESX genes are among those preferentially required in Rag1-/- (though obviously they are also required in B6, so presumably this does not account for the 'additional' genes required in Rag1-/-). Could the authors elaborate a bit more on what they think is going on here?

Reviewer 1 also requested additional discussion of gene set sizes and functional insights. While we hesitate to speculate in too much detail, additional details are now included in the discussion (lines 592-599).

Line 281: "As more CC strains, and presumably more distinct immune states, were included in the analysis, the cumulative number of genes necessary for growth in these animals also increased". It is possible that repeating the experiment with more and more B6 animals would also increase the cumulative number of genes necessary? In other words, might the sensitivity of the TnSeq method be what is limiting here as opposed to host genetic diversity? I presume the authors have a good argument against this concern; it would be helpful for readers to hear it.

This was a very helpful suggestion. The revised manuscript includes a study verifying that the inclusion of additional B6 libraries does not substantially increase the estimate of genes necessary for growth. For this study, we randomly paired six replicate TnSeq libraries from 4-week post infection B6 mice to identify genes required for growth in each of the three runs vs cumulatively. The process was repeated 10 times with the results shown in Figure4-FigureSupplement 1 and show that a single “run”, which is equivalent to the data generated from each genotype in our larger study, was sufficient to identify the majority of genes, and adding additional datasets did not substantially increase the cumulative gene count.

Line 295: "the resulting mini-pool was subjected to in vivo selection in the same manner as the TnSeq study" – does this mean that the minipool was used to infect ALL the same mouse strains (all the CC mice) as in Figure 4A? This isn't clear. A related concern regarding Figure 4C: why are selected mouse strains shown in this figure? Are they the strains that showed the biggest signal in the TnSeq experiment? The worry is they were cherry picked. This could be explained better.

A small focused pool of bacterial mutants was used to infect a subset of CC mice, which were chosen based on TnSeq data. Thus, this experiment was designed to test a number of individual observations from the TnSeq dataset, and these are shown in the figure. We have added additional explanation of our rationale (lines 309-319). Importantly, we note that, “data from all reliably-detected strains is shown in Figure 4C” to alleviate the reviewer’s concern that the presented data were “cherry picked”.

Line 298: "relative abundance predicted by TnSeq was reproduced with deletion mutants" – this is difficult to evaluate because the fitness of each of the mutants in each of the strains in Figure 4A is not stated. I realize the full dataset of all the TnSeq data is available as a spreadsheet but it would be nice if the reader didn't have to dig through that. Ideally it would be great to create a searchable website in which investigators could enter their gene of interest and the strains in which the mutant shows a phenotype (and the magnitude of the phenotype) could be displayed.

Both the TnSeq data and the corresponding data from the minipool infection is provided in Figure 4C, so it should not be necessary for the reader to refer to the supplementary tables. The suggested searchable database including these data and many other publicly available datasets has been constructed, and will be reported in a separate publication.

Line 312: again, it is not clear what the TnSeq predicted. Please state that result here too.

Done (line 342-344).

Line 435: which HipQTL are associated with both ahpC and eccD1? Please specify. I don't see this information in the cited figure either (Figure 4E). Please make sure this is correct.

Done (line 457-458).

Reviewer #4:I reviewed this study only to evaluate the robustness of the results and the statistical evaluations used in the study. No attention was paid to novelty or scientific interest. Some of the statistical approaches used by the authors were beyond my expertise and require further analysis by a professional statistician to which I lay no claim to being. This study used multiple simultaneous comparisons of both animal strains and bacterial strains, with what appears to be generally quite small sample size, non-replicated experiments.

The revised manuscript contains independent validation of the susceptibility and cytokine production traits in Figure 1 (including Figure1-SourceData2), independent validation of an important QTL in Figure 3 (including Figure3-SourceData1 and Figure3-SourceData2), and independent validation of the TnSeq predictions in Figure 4 (including Figure4-SourceData2). In all cases, our larger datasets have proven to be robust and reproducible.

Data files that are presented either omit entirely some experiments or provide only summary data of mean values without indicating numbers of subjects, measures of variability such as standard deviation; no information on the results of replicate studies, if any were performed, are in these data files. Therefore I find it impossible to determine if many of the "significant" associations are robust.

Several of the reviewer’s specific examples were quite helpful, and we respond to each below.

1) Data for Figure 1. The data set for figure 1 only summary data. There is no indication of the number of animals studied in each group and no indication of standard deviation. Also, there is no indication in the data set or in the text if any of the experimental results for 1A to 1D were based on replicate studies. The underlying data for figures E to H are not available in the figure 1 data file; the text gives the range of subject sizes and states that two independent experiments were performed, but it is unclear if the data presented in the figures represents both experiments, just one experiment, or if the data presented includes both experiments whether the results were simply combined or if there was some normalization performed. The statistical analysis method does not say whether the data were log-transformed before analysis. All of the author's conclusions may be absolutely correct but it is difficult to know this without more detailed information.

The reviewer highlights a significant oversight – the lack of details in our supplementary table. This has been remedied. We now provide the group size, mean, and standard deviation for each measurement. We have deposited all raw data in our Sassetti Lab GitHub repository, which is publicly available. The other comments refer to data that has been removed in response to Reviewer 1.

2) Data for Figure 4. The data set for this figure only includes the underlying data for 4A. No underlying data are given for 4C, D and E. No indication is given of the number of repeated experiments, and if these were performed what the results were. No method is given for the statistical analysis used in 4D. The authors conclude that there are significant differences between mouse strains for several Mtb strains, but give no information on the numbers of mice studied, variation within groups, repeat experiments, etc; also no statistical analysis is performed. Again, the author's conclusions may be absolutely correct but it is difficult to know this without seeing the underlying data.

The underlying data for 4A was all included in the previous submission (Figure4-SourceData1). This is also the source data for Figure 4E as mentioned in the figure legend (line 967). For 4C, all mouse numbers were in the legend (line 955) and we additionally include source data file of counts for each Mtb mutant in individual mice (Figure4-SourceData2). For 4D, the number of mice were in the legend (line 960) and we have added additional text to specify the statistical testing conducted between the lung burden of B6 and CAST mice at each specified timepoint (lines 960-961).

While the authors may believe that their answers to the transparent reporting form were complete and accurate, based on what I have found I'm not certain that this is correct.

We verify that the revised manuscript complies with the transparent reporting requirements.

Reviewer #5:Sample size and reproducibility. The groups were small (average of 3 mice, sometimes only 2), which could result in a high variance per group. It would be important to evaluate how reproducible are the phenotypic measurements per strain. The authors mention reproducibility per genotype (Page 8 line 177), but they do not clearly show it.

The reviewer refers to our heritability analysis which was provided in Figure1-FigureSupp3. Heritability is the fraction of variation attributable to between strain differences relative to within strain noise. This is explicitly: SS(strain)/SS(total) in an ANOVA table (where SS(total) is SS(strain)+SS(error)) (SS; sum of squares). Our heritability analysis shows that >75% of variation could be attributed to genetic differences for the majority of traits. In the revised manuscript, we also provide p-values of each trait showing that genotype has a significant effect on all traits.

In addition, the revised manuscript includes group size, mean, and standard deviation for each trait and genotype in Figure1-SourceData1, validation of susceptibility and cytokine abundance traits in Figure 1 (Figure 1 E-H and Figure1-SourceData2), validation of an important QTL in Figure 3C (Figure3-SourceData1 and 2), and validation of TnSeq measurements in Figure 4 (Figure4-SourceData2). This provides a transparent assessment of technical variability, and demonstrates the reproducibility of our dataset.

Perhaps a dimension reduction approach (PCA or UMAP) could be used to evaluate the distance between replicates.

This is a very interesting idea, and we appreciate the suggestion. However, it is difficult to envision how PCA or UMAP could produce a quantitative assessment of genotype-dependent variation that is more robust than the heritability analysis we provide in Table S2.

As specific examples: How many animals were used per group in outlier strains (Figure 1 C and D)? What was the variance for the y-axis measurements? Could variance explain the outlier nature of the strain?

Outliers were identified by linear regression using studentized residuals, which accounts for intra-genotype variation. This is now explained more clearly in the main text (lines 179-183) and figure legend.

QTL analysis. The CC QTL analysis faces similar challenges to genetic studies in humans. Overlooking complex population structure can lead to inflated false positives but can also prevent harvesting the full potential of the samples. Limitations of the QTL analysis are briefly introduced (Page 11 line 241) but the explanation in methods is convoluted and difficult to follow. The study could benefit from using their SNPs data for genome-wide association testing using all strains correcting by population structure. This strategy was shown to be robust when performed to a study of similar framework using different inbred strains and their crosses (PMID 32365090). This method could address sample size and QTL as the animals would be treated as unique samples (not per group) correcting by population structure (similarities between samples in the dataset). If correct, this would be a more straightforward genome-wide analysis.

We appreciate the reviewer’s constructive suggestions. While the reviewer is entirely correct that in certain situations (such as the cited study, PMID 32365090), direct SNP-based approaches can allow for improved resolution of causal variants, we would argue that in many cases these SNP approaches would in fact further cloud the analyses.

The field standard for CC studies (Aylor, Valdar et al. 2011, Ferris, Aylor et al. 2013, Gralinski, Ferris et al. 2015, Keele, Crouse et al. 2019) has been the haplotype-based genetic mapping which also includes a kinship (or overall relatedness) correction (Churchill, Gatti et al. 2012, Broman, Gatti et al. 2019). This is largely due to the fact that (as shown in(Aylor, Valdar et al. 2011), in a finite population such as the CC, the strain distribution patterns of single SNPs can lead to false associations (due to long-range linkage disequilibrium). In the cited case, a simple monogenic trait (the white spot) was assessed, and the problem should only be exacerbated for multi-genic traits).

Furthermore, we have strong evidence presented here that several of our discovered QTL are due to multiple alleles (i.e. we find haplotypes associated with high, medium and low phenotypes). Members of our team have shown this causally before (Ferris, Aylor et al. 2013). As any given SNP cannot fully capture 3 (or more) states, the field has acknowledged that haplotype mapping is the state of the art.

We agree that in the future, with designs like the DO studies (Chick, Munger et al. 2016, Keele, Zhang et al. 2021), or F2 crosses (Smith, Proulx et al. 2019), will facilitate improved SNP (or copy + structural variant) mapping and we are excited to begin anticipating these.

Detailed comments:In methods, it is stated that TipQTL significance was assessed using permutations. However, in the Results section and in table 1 it is not clearly stated if the p values are raw or corrected via the permutation.

This has been corrected.

Line 63: Co-existence of Mtb and *Homo sapiens* is now widely accepted, perhaps 2000-6000 years. Has co-evolution been demonstrated or is this hypothesized? Same point with the role of genetic variation presented as established.

Additional citations have been added (lines 63-64).

When describing correlations, suggest a standard terminology: Strong, Moderate, Weak. Highly correlated is of uncertain meaning. On line 182 the R2 was 0.57 but the text said correlated. Some use the following: 0-0.19 = very weak, 0.2-0.39 = weak, 0.40-0.59 = moderate, 0.6-0.79 = strong, 0.8-1 = very strong correlation

This is an excellent idea, and we edited the text accordingly.

Line 151. What does it mean when the susceptibility is largely consistent with other studies. Can the authors spell out the inconsistencies somehow?

This text was edited to be more explicit (lines 151-155).

Line 220. What is reasonable statistical confidence?

The text was edited to remove this confusing reference.

Line 272. At some point, it would be useful to read in the text that the combined studies of WT Bl6, RAG^-/-^, IFN-/-, Nos2-/- and CYbb-/- mice produced x essential genes, and that the CC raised it from x to y. The paragraph tells us the 234 genes were identified in Bl6 and that the plateau was ~750. How far from 234 to 750 was reached with the 4 specified knockout mice?

The requested statement was added (lines 302-306).

Line 321. I did not understand what was complex about the kinetics. Perhaps this can be clarified?

This text was clarified (Lines 342-344).

References:

Aylor, D. L., W. Valdar, W. Foulds-Mathes, R. J. Buus, R. A. Verdugo, R. S. Baric, M. T. Ferris, J. A. Frelinger, M. Heise, M. B. Frieman, L. E. Gralinski, T. A. Bell, J. D. Didion, K. Hua, D. L. Nehrenberg, C. L. Powell, J. Steigerwalt, Y. Xie, S. N. Kelada, F. S. Collins, I. V. Yang, D. A. Schwartz, L. A. Branstetter, E. J. Chesler, D. R. Miller, J. Spence, E. Y. Liu, L. McMillan, A. Sarkar, J. Wang, W. Wang, Q. Zhang, K. W. Broman, R. Korstanje, C. Durrant, R. Mott, F. A. Iraqi, D. Pomp, D. Threadgill, F. P. de Villena and G. A. Churchill (2011). "Genetic analysis of complex traits in the emerging Collaborative Cross." Genome Res 21(8): 1213-1222.

Broman, K. W., D. M. Gatti, P. Simecek, N. A. Furlotte, P. Prins, Ś. Sen, B. S. Yandell and G. A. Churchill (2019). "R/qtl2: Software for Mapping Quantitative Trait Loci with High-Dimensional Data and Multiparent Populations." Genetics 211(2): 495-502.

Chick, J. M., S. C. Munger, P. Simecek, E. L. Huttlin, K. Choi, D. M. Gatti, N. Raghupathy, K. L. Svenson, G. A. Churchill and S. P. Gygi (2016). "Defining the consequences of genetic variation on a proteome-wide scale." Nature 534(7608): 500-505.

Churchill, G. A., D. C. Airey, H. Allayee, J. M. Angel, A. D. Attie, J. Beatty, W. D. Beavis, J. K. Belknap, B. Bennett, W. Berrettini, A. Bleich, M. Bogue, K. W. Broman, K. J. Buck, E. Buckler, M. Burmeister, E. J. Chesler, J. M. Cheverud, S. Clapcote, M. N. Cook, R. D. Cox, J. C. Crabbe, W. E. Crusio, A. Darvasi, C. F. Deschepper, R. W. Doerge, C. R. Farber, J. Forejt, D. Gaile, S. J. Garlow, H. Geiger, H. Gershenfeld, T. Gordon, J. Gu, W. Gu, G. de Haan, N. L. Hayes, C. Heller, H. Himmelbauer, R. Hitzemann, K. Hunter, H. C. Hsu, F. A. Iraqi, B. Ivandic, H. J. Jacob, R. C. Jansen, K. J. Jepsen, D. K. Johnson, T. E. Johnson, G. Kempermann, C. Kendziorski, M. Kotb, R. F. Kooy, B. Llamas, F. Lammert, J. M. Lassalle, P. R. Lowenstein, L. Lu, A. Lusis, K. F. Manly, R. Marcucio, D. Matthews, J. F. Medrano, D. R. Miller, G. Mittleman, B. A. Mock, J. S. Mogil, X. Montagutelli, G. Morahan, D. G. Morris, R. Mott, J. H. Nadeau, H. Nagase, R. S. Nowakowski, B. F. O'Hara, A. V. Osadchuk, G. P. Page, B. Paigen, K. Paigen, A. A. Palmer, H. J. Pan, L. Peltonen-Palotie, J. Peirce, D. Pomp, M. Pravenec, D. R. Prows, Z. Qi, R. H. Reeves, J. Roder, G. D. Rosen, E. E. Schadt, L. C. Schalkwyk, Z. Seltzer, K. Shimomura, S. Shou, M. J. Sillanpää, L. D. Siracusa, H. W. Snoeck, J. L. Spearow, K. Svenson, L. M. Tarantino, D. Threadgill, L. A. Toth, W. Valdar, F. P. de Villena, C. Warden, S. Whatley, R. W. Williams, T. Wiltshire, N. Yi, D. Zhang, M. Zhang and F. Zou (2004). "The Collaborative Cross, a community resource for the genetic analysis of complex traits." Nat Genet 36(11): 1133-1137.

Churchill, G. A., D. M. Gatti, S. C. Munger and K. L. Svenson (2012). "The Diversity Outbred mouse population." Mamm Genome 23(9-10): 713-718.

Ferris, M. T., D. L. Aylor, D. Bottomly, A. C. Whitmore, L. D. Aicher, T. A. Bell, B. Bradel-Tretheway, J. T. Bryan, R. J. Buus, L. E. Gralinski, B. L. Haagmans, L. McMillan, D. R. Miller, E. Rosenzweig, W. Valdar, J. Wang, G. A. Churchill, D. W. Threadgill, S. K. McWeeney, M. G. Katze, F. Pardo-Manuel de Villena, R. S. Baric and M. T. Heise (2013). "Modeling host genetic regulation of influenza pathogenesis in the collaborative cross." PLoS Pathog 9(2): e1003196.

Graham, J. B., J. L. Swarts, M. Mooney, G. Choonoo, S. Jeng, D. R. Miller, M. T. Ferris, S. McWeeney and J. M. Lund (2017). "Extensive Homeostatic T Cell Phenotypic Variation within the Collaborative Cross." Cell Rep 21(8): 2313-2325.

Gralinski, L. E., M. T. Ferris, D. L. Aylor, A. C. Whitmore, R. Green, M. B. Frieman, D. Deming, V. D. Menachery, D. R. Miller, R. J. Buus, T. A. Bell, G. A. Churchill, D. W. Threadgill, M. G. Katze, L. McMillan, W. Valdar, M. T. Heise, F. Pardo-Manuel de Villena and R. S. Baric (2015). "Genome Wide Identification of SARS-CoV Susceptibility Loci Using the Collaborative Cross." PLoS Genet 11(10): e1005504.

Keele, G. R., W. L. Crouse, S. N. P. Kelada and W. Valdar (2019). "Determinants of QTL Mapping Power in the Realized Collaborative Cross." G3 (Bethesda) 9(5): 1707-1727.

Keele, G. R., T. Zhang, D. T. Pham, M. Vincent, T. A. Bell, P. Hock, G. D. Shaw, J. A. Paulo, S. C. Munger, F. Pardo-Manuel de Villena, M. T. Ferris, S. P. Gygi and G. A. Churchill (2021). "Regulation of protein abundance in genetically diverse mouse populations." Cell Genomics: 100003.

Lorè, N. I., B. Sipione, G. He, L. J. Strug, H. J. Atamni, A. Dorman, R. Mott, F. A. Iraqi and A. Bragonzi (2020). "Collaborative Cross Mice Yield Genetic Modifiers for *Pseudomonas aeruginosa* Infection in Human Lung Disease." mBio 11(2).

Minato, Y., D. M. Gohl, J. M. Thiede, J. M. Chacon, W. R. Harcombe, F. Maruyama and A. D. Baughn (2019). "Genomewide Assessment of *Mycobacterium tuberculosis* Conditionally Essential Metabolic Pathways." mSystems 4(4).

Mishra, B. B., R. R. Lovewell, A. J. Olive, G. Zhang, W. Wang, E. Eugenin, C. M. Smith, J. Y. Phuah, J. E. Long, M. L. Dubuke, S. G. Palace, J. D. Goguen, R. E. Baker, S. Nambi, R. Mishra, M. G. Booty, C. E. Baer, S. A. Shaffer, V. Dartois, B. A. McCormick, X. Chen and C. M. Sassetti (2017). "Nitric oxide prevents a pathogen-permissive granulocytic inflammation during tuberculosis." Nat Microbiol 2: 17072.

Noll, K. E., A. C. Whitmore, A. West, M. K. McCarthy, C. R. Morrison, K. S. Plante, B. K. Hampton, H. Kollmus, C. Pilzner, S. R. Leist, L. E. Gralinski, V. D. Menachery, A. Schäfer, D. Miller, G. Shaw, M. Mooney, S. McWeeney, F. Pardo-Manuel de Villena, K. Schughart, T. E. Morrison, R. S. Baric, M. T. Ferris and M. T. Heise (2020). "Complex Genetic Architecture Underlies Regulation of Influenza-A-Virus-Specific Antibody Responses in the Collaborative Cross." Cell Rep 31(4): 107587.

Sakai, S., K. D. Kauffman, M. A. Sallin, A. H. Sharpe, H. A. Young, V. V. Ganusov and D. L. Barber (2016). "CD4 T Cell-Derived IFN-γ Plays a Minimal Role in Control of Pulmonary *Mycobacterium tuberculosis* Infection and Must Be Actively Repressed by PD-1 to Prevent Lethal Disease." PLoS Pathog 12(5): e1005667.

Smith, C. M., M. K. Proulx, R. Lai, M. C. Kiritsy, T. A. Bell, P. Hock, F. Pardo-Manuel de Villena, M. T. Ferris, R. E. Baker, S. M. Behar and C. M. Sassetti (2019). "Functionally Overlapping Variants Control Tuberculosis Susceptibility in Collaborative Cross Mice." mBio 10(6).

[Editors’ note: what follows is the authors’ response to the second round of review.]

Reviewers and I concur that your manuscript provides value to the field as a "Tools and Resource" paper for generating new hypotheses. The genotypic diversity explored in the study, together with an assessment of disease pathology/outcomes, provides a useful framework to pose important questions that are currently topical for tuberculosis. That said, it would be important to frame the study as a resource rather than trying to probe specific questions with the current dataset, which can be limited by the depth of data available currently in the work. I encourage you to consider carefully how best to present this to the field in a manner that acknowledges the limitations and but also creates the space for innovative and explorative future work.

We thank the reviewers and editor for review of our work. To facilitate the paper as a useful “Tools and Resources” paper, we have added text throughout the manuscript highlighting the specific Tools and Resources aspects of each section, detailed the limitations and discussed further avenues for follow-up and new lines of enquiry this work has opened.

To facilitate this, please address the following concerns:1. For a paper reporting a potentially useful resource, it is imperative that all data be publicly available for further interrogation and exploration. Reviewers expressed concern that this is not the case with your submission. For the GEO link, data appear available at https://www.ncbi.nlm.nih.gov/geo/query/acc.cgi?acc=GSE164156. However, for the GitHub link, the Smith repository was not located. There was only one repository (Bellerose): https://github.com/sassettilab?tab=repositories. Please address this.

This GitHub repo has been made public.

2. In lines 145-147, it is stated: Groups of 3-6 male mice per genotype were infected and TB disease-related traits were quantified at one-month post-infection. Data from all surviving animals are provided in Figure 1-sourcedata1. However, the source data file indicates that the N is 1-3 for most of the data, except for two strains, Bl6 and CC010, which are reported to have 6 mice. There are asterisks for significance in only in a minority of the bars (~15) in Figure 1A. What about the remainder? What n's did they have? Too few to determine significance? This should be clarified.

We clarified this after previous review to indicate that while at least 3 mice of each genotype was infected, some individual mice within a group died early and hence were not phenotyped. To make this even more explicit, we have added a column into Figure1-SourceData1_CCTB_Phenotypes to state “N of mice infected” and have changed the “N” column to “N of surviving phenotyped animals”.

3. Explain why an R value of 0.43 would be considered a moderate correlation, when elsewhere you suggest that there could be biological differences that may make this correlation weak. Similarly, it is often the case that in inflammatory processes, TNF and IFNγ track together, even if made by different cells, which is not even strictly the case. These are two examples of continued disagreement. Please reconsider carefully how you present these data.

This standardized nomenclature is defined in the text and was used in response to the previous review. Since the interpretation of this terminology is subjective, we use a systematic nomenclature.

4. Line 184-188. The "significant" associations described in this section are for non-replicate studies with N=2 (CC30, 40 and A/J). Perhaps reconsider the use of "significant" in this context. In Line 253, this experiment used a large N and hence likely produced robust results.

The word “significant” has been removed.

5. Figure 1 A. Giving results of an ANOVA with post-hoc analysis comparison of means to a single mouse strain (C57/BL6) is difficult to interpret because of the potential bias from post-hoc decisions about which strain serves as a comparator. For example, using AJ as a comparator shows no significant differences with other strains. For this reason, excluding this analysis is warranted.

C57BL/6J was used as the comparator as that is the standard mouse strain used in the TB field. As a Tools and Resources paper, comparing this new mouse panel to the accepted/standard mouse strain C57BL/6J is the comparisons most researchers immediately would look for (and have enquired about from the preprint) and thus would be most beneficial for researchers in the field. A sentence has been added to figure legend 1 to explicitly point researchers to the most useful comparisons.

6. Line 199: "A number of genotypes were able to control bacterial replication yet had very low levels of IFNΓ (Figure 1D)" -- for the benefit of the reader, please specify which genotypes fall into this category. Ideally this would be specifically mentioned in the text and in addition the genotypes of interest would be labelled in the figure. Similarly for Figure 1C, it would be helpful if the significant "outlier" genotypes were labelled in the figure so the reader can know which these are.

These genotypes have been specified again in the text and figure legend.

7. Line 487: “Two TipQTL overlapped with HipQTL (Figure 7)…”. Please specify in the text which overlapping QTL are being referred to here. It seems evident from the figure but specifying would assist the reader here.

This has been specified in the text (line 492 and highlighted in the marked up manuscript).

8. Some of the type on the figures is really tiny, e.g., 1A, 1B, most of figure 4, parts of figure 6, etc. Enlarge the panels with the tiniest type so that they are more readable.

All figures have been enlarged to the maximum size permittable.